# How Many Domains Suffice for Domain Generalization? A Tight Characterization via the Domain Shattering Dimension

**Cynthia Dwork**
Harvard University
dwork@seas.harvard.edu

**Lunjia Hu** *
Northeastern University
lunjia@alumni.stanford.edu

**Han Shao** †
University of Maryland, College Park
hanshao@umd.edu

## Abstract

We study a fundamental question of domain generalization: given a family of domains (*i.e.*, data distributions), how many randomly sampled domains do we need to collect data from in order to learn a model that performs reasonably well on every seen and unseen domain in the family? We model this problem in the PAC framework and introduce a new combinatorial measure, which we call the *domain shattering dimension*. We show that this dimension characterizes the *domain sample complexity*. Furthermore, we establish a tight quantitative relationship between the domain shattering dimension and the classic VC dimension, demonstrating that every hypothesis class that is learnable in the standard PAC setting is also learnable in our setting.

## 1 Introduction

The ability to generalize across domains is an important component of human intelligence and a crucial milestone in the development of increasingly powerful artificial intelligence. It is not surprising that an experienced driver in one country can reasonably, albeit imperfectly, drive in another country even without further training. It is also not surprising that a skilled chess player can outperform an average person on a totally different board game. An expert doctor who has studied a disease using data from a few large hospitals can potentially provide reasonable treatment for patients in a geographically remote and biologically different population that does not have access to those large hospitals and has not been the subject of prior study. Such domain generalization abilities are empowered by the capability of the learner (i.e., the driver, chess player, or doctor) to distill and master universal laws (about driving, game playing, or medical treatment) that hold even on unseen domains, while separating them from idiosyncratic patterns that are domain-specific.

In this work, we study the theoretical foundations of domain generalization with a focus on capturing this ability of learning "universal laws" while not overfitting to the "idiosyncratic patterns". Consider a family $\mathcal{G}$ of domains, where every domain $\mathcal{D}$ is a distribution on examples $(x, y)$ each consisting of an instance $x \in X$ and a binary label $y \in \{0, 1\}$. On a specific domain $\mathcal{D}$, there may be a hypothesis

---

*Work done while at Harvard. Supported by the Simons Foundation Collaboration on the Theory of Algorithmic Fairness and the Harvard Center for Research on Computation and Society (CRCS).

†Work done while at Harvard. Supported by the Harvard Center of Mathematical Sciences and Applications (CMSA).

(i.e., classifier) $h : X \to \{0, 1\}$ with low classification error $\mathrm{err}_{\mathcal{D}}(h)$, say, below 0.01:

$$\mathrm{err}_{\mathcal{D}}(h) := \Pr_{(x,y) \sim \mathcal{D}}[y \neq h(x)] \leq 0.01.$$

However, the strong performance of this hypothesis may rely on patterns specific to the current domain and may fail to transfer to other domains $\mathcal{D}' \in \mathcal{G}$. For the purpose of domain generalization, we would instead prefer a different hypothesis $h^\star$ that achieves reasonably low—though not necessarily minimal—error across *all* domains $\mathcal{D} \in \mathcal{G}$. Specifically, for a mild but meaningful error threshold, say 0.3, we assume

$$\max_{\mathcal{D} \in \mathcal{G}} \mathrm{err}_{\mathcal{D}}(h^\star) \leq 0.3. \tag{1}$$

Motivated by this, we model the underlying "universal laws" by assuming the existence of such a universally good hypothesis $h^* \in \mathcal{H}$ that satisfies Equation (1). We make no further assumptions on the structure of the domains in $\mathcal{G}$ or on the relationships among them.

In our domain generalization task, a learner has data access to a limited number of domains $\mathcal{D}_1, \ldots, \mathcal{D}_k$ sampled i.i.d. from a meta-distribution $\mathcal{P}$ over domain space $\mathcal{G}$. Using this limited data access, the learner's goal is to output a model $h : X \to \{0, 1\}$ that achieves reasonably good performance on new, unseen domains drawn from $\mathcal{P}$ *without any additional training*. Concretely, for performance threshold and error parameter $\tau, \gamma \in [0, 1]$, the learner's goal is to output $h : X \to \{0, 1\}$ such that

$$\Pr_{\mathcal{D} \sim \mathcal{P}}[\mathrm{err}_{\mathcal{D}}(h) \leq \tau] \geq 1 - \gamma.$$

In this work, we aim to answer the following quantitative question:

**How many domains does the learner need to see to achieve domain generalization?**

The answer to this question, which we call the *domain sample complexity*, is analogous to the notion of sample complexity in classic learning theory. Sample complexity is defined as the number of *data points* needed to learn a good model on a *single* domain, whereas our notion of domain sample complexity measures the number of *domains* the learner needs to collect data from in order to generalize to unseen domains. Thus, domain sample complexity can be viewed as a higher-level meta-variant of the notion of sample complexity.

Characterizing the sample complexity of various learning tasks is a central question in learning theory. For example, the celebrated VC theory characterizes the sample complexity of classic PAC learning [Valiant, 1984] using the VC dimension [Vapnik and Chervonenkis, 1971], a combinatorial complexity measure of the hypothesis class $\mathcal{H}$. The goal of our work is to characterize the domain sample complexity, which depends not only on the complexity of the hypothesis class $\mathcal{H}$, but also on the *interaction* between $\mathcal{H}$ and the domain family $\mathcal{G}$. For example, even if $\mathcal{H}$ is itself very complex, the domain sample complexity can still be very small if the domains in $\mathcal{G}$ are very similar. Even when both $\mathcal{H}$ and $\mathcal{G}$ are complex, the domain sample complexity can still be small if their complexities are concentrated on disjoint subsets of the input space $X$. For instance, suppose all hypotheses $h \in \mathcal{H}$ are identical on a subset $X_1 \subseteq X$, and $\mathcal{G}$ only contains domains $\mathcal{D}$ satisfying $\Pr_{\mathcal{D}}[x \in X_1] = 1$. In this case $\mathcal{H}$ can be complex outside of $X_1$ and $\mathcal{G}$ can be complex inside $X_1$, but the domain sample complexity is always zero: on every fixed domain $\mathcal{D} \in \mathcal{G}$, all hypotheses $h \in \mathcal{H}$ achieve the same error. Finding a combinatorial quantity that tightly characterizes the domain sample complexity requires accurately capturing this interaction between $\mathcal{H}$ and $\mathcal{G}$.[3]

## 1.1  Our Contributions

We model the domain generalization problem in the PAC framework and introduce a new combinatorial measure, the *domain shattering dimension* (Definition 4.1), for a hypothesis class $\mathcal{H}$ with respect to a family $\mathcal{G}$ of domains.

For each hypothesis $h$, we define the function $\mathrm{err}_{\cdot}(h) : \mathcal{G} \to \mathbb{R}$ that maps each domain to the error of $h$ on that domain, thereby inducing a new function class. A natural idea is to analyze the fat-shattering dimension [Kearns and Schapire, 1994] of this class. This turns out to overestimate the domain sample complexity. For a target error threshold $\tau$ (e.g. $\tau = 0.3$), the domain sample complexity can

---

[3]A similar situation that requires capturing the *interaction* between a pair of hypothesis classes occurs in the study of *comparative learning* (hybrid of *realizable* and *agnostic* learning) [Hu and Peale, 2023, Hu et al., 2022].

be very small (e.g. when $\mathrm{err}_\mathcal{D}(h) \leq \tau$ is achieved by every $h \in \mathcal{H}$ on every domain $\mathcal{D} \in \mathcal{G}$), but the size of a shattered set can be large at a different threshold $\tau'$, causing the fat-shattering dimension to be large as well.

To address this, we propose a new dimension by modifying the fat-shattering dimension: instead of allowing different thresholds across shattered domains, we require a uniform and fixed threshold across all of them. We show that the resulting domain shattering dimension gives both upper (Theorem 4.1) and lower (Theorem 4.3) bounds on the domain sample complexity, and these bounds match up to a poly-logarithmic factor. We obtain the upper bound by a min-max variant of the empirical risk minimization (ERM) algorithm. To analyze the domain sample complexity achieved by the min-max ERM algorithm, we establish a uniform convergence bound (Lemma 4.2) for *partial concept classes* based on a generalized Sauer-Shelah-Perles lemma by Alon et al. [2022], which may be of independent interest. Our algorithm and upper bound can be directly applied to other learning tasks beyond binary classification, such as multi-class classification and regression (Remark 4.2).

The next question we address in this paper is *the relationship between domain generalization learnability and standard PAC learnability*. It is well known that PAC learnability is characterized by the VC dimension. We compare our domain shattering dimension with the VC dimension and show that, for a hypothesis class $\mathcal{H}$ with VC dimension $d$ and a margin $\alpha$, the domain shattering dimension is always upper bounded by $O(d \log(1/\alpha))$ (Theorem 5.2). Moreover, we construct a hypothesis class with VC dimension $d$ and a domain family $\mathcal{G}$ for which the domain shattering dimension is $\Omega(d \log(1/\alpha))$ (Theorem 5.1). This establishes a tight relationship between the two measures and demonstrates that standard PAC learnability implies domain generalization learnability, but the domain sample complexity can be much smaller than the sample complexity for PAC learning.

Finally, we relate our work to the literature on domain adaptation. In domain adaptation, the goal is to generalize from a source distribution to a related target distribution, under the assumption that the two are sufficiently similar. This similarity is often quantified using measures such as the $\mathcal{H}$-divergence introduced in Ben-David et al. [2010]. We show that if all domains in $\mathcal{G}$ are similar to each other under a metric based on a modified version of the $\mathcal{H}$-divergence, then the domain shattering dimension is $1$. Moreover, when the hypothesis class admits a finite cover under this metric, the domain shattering dimension can be upper bounded by the covering number (Theorem 6.1). These results illustrate the potential of our domain shattering dimension as a general notion for characterizing domain generalization without explicitly modeling how domains are related.

## 1.2 Related Work

There is a vast literature on domain generalization and related learning paradigms such as meta-learning, zero-shot learning, domain adaptation, out of distribution generalization, transfer learning, invariant risk minimization, and multi-task learning. The excellent survey of domain generalization of Wang et al. [2022] provides a helpful taxonomy of many of these paradigms and explores theoretical underpinnings of domain adaptation and domain generalization. Zhou et al. [2022] organizes a plethora of works by (1) application area; (2) method; and (3) learning paradigm, and provides pointers to many datasets commonly used for domain generalization.

Our work is distinguished from prior theoretical work in that simultaneously (1) we make only the minimalist assumption of the existence of a universally good hypothesis $h^* \in \mathcal{H}$, (e.g., one satisfying Equation (1)), with no further requirements regarding how different domains in $\mathcal{G}$ are related, making our model very general; and (2) we obtain *provable* results for the strong learning objective of requiring the output model $h$ to achieve a reasonably low error simultaneously on essentially *all* domains (including the unobserved ones), rather than achieving low error *in expectation* over choice of domain. Achieving low error simultaneously on all domains can be substantially different from achieving low average error when $\tau$ is not very close to zero. For example, an average error of $0.25$ could arise if the hypothesis incurs zero error on $3/4$ of the domains but makes completely incorrect predictions (error $1$) on the remaining $1/4$ of the domains. Finally, we explicitly focus on the question of *domain* sample complexity – the total number of domains that need to be sampled in order to generalize well to a random new domain. This is different from the common notions of sample complexity and query complexity, and this lens yielded the key concept of the domain shattering dimension.

## 2 Preliminaries

We include the definition of *partial concepts* that will be very useful in our analysis. A partial concept is a function $f : Z \to \{0, 1, \bot\}$ that assigns each individual $z$ in an input space $Z$ a binary label (0 or 1), or the "unknown" label, denoted by $\bot$. A *total* concept $f : Z \to \{0, 1\}$ assigns a binary label (0 or 1) to each individual $z \in Z$ without using the "unknown" label $\bot$. Throughout the paper, for a positive integer $n$, we use $[n]$ to denote the set $\{1, \ldots, n\}$.

**Definition 2.1** (VC dimension of partial concept classes [Vapnik and Chervonenkis, 1971, Alon et al., 2022]). *Let $\mathcal{F}$ be a class of partial concepts $f : Z \to \{0, 1, \bot\}$ on an arbitrary input space $Z$. We say a subset $S \subseteq Z$ is* shattered *by $\mathcal{F}$ if for every subset $E \subseteq S$, there exists $f_E \in \mathcal{F}$ such that*

$$f_E(s) = 0 \quad \text{for every } s \in E,$$
$$f_E(s) = 1 \quad \text{for every } s \in S \setminus E.$$

*The VC dimension of $\mathcal{F}$ (denoted $\mathrm{VCdim}(\mathcal{F})$) is the size of the largest shattered set $S \subseteq Z$.*

Clearly, the classical VC dimension, defined for total concept classes, is a special case of the VC dimension for partial concept classes.

## 3 Problem Setup for Domain Generalization

Given input space $X$ and label space $Y = \{0, 1\}$, a domain (or data distribution) $\mathcal{D}$ is a distribution over $X \times Y$. We consider a family $\mathcal{G}$ of domains $\mathcal{D}$. For any hypothesis/predictor $h : X \to Y$, we define the error rate of $h$ under domain $\mathcal{D}$ as

$$\mathrm{err}_{\mathcal{D}}(h) := \mathbb{P}_{(x,y) \sim \mathcal{D}}(y \neq h(x)).$$

We consider an underlying meta distribution $\mathcal{P}$ over the domains $\mathcal{D} \in \mathcal{G}$. Given a threshold $\tau$, for any hypothesis $h$ and a threshold $\tau$, we define the domain error of $h$ with respect to $\tau$ as

$$\mathrm{Er}_{\mathcal{P}, \tau}(h) := \mathbb{P}_{\mathcal{D} \sim \mathcal{P}}(\mathrm{err}_{\mathcal{D}}(h) > \tau),$$

which quantifies the probability mass of the domains where $h$ incurs error greater than $\tau$. It follows immediately that $\mathrm{Er}_{\mathcal{P}, \tau}(h)$ is monotonically decreasing in $\tau$.

As in the standard PAC learning setting, we are given a hypothesis class $\mathcal{H}$ and aim to output a hypothesis that performs well compared to the best hypothesis in $\mathcal{H}$.

**Definition 3.1** (Optimal domain error bound and optimal hypothesis). *Given a hypothesis class $\mathcal{H}$, domain class $\mathcal{G}$, and a distribution $\mathcal{P}$ over $\mathcal{G}$, the optimal domain error bound is defined as*

$$\tau^{\star}_{\mathcal{P}, \mathcal{H}} = \min\{\tau | \exists h \in \mathcal{H}, \mathrm{Er}_{\mathcal{P}, \tau}(h) = 0\},$$

*and the optimal hypothesis $h^{\star}$ is defined as one hypothesis $h$ achieving $\mathrm{Er}_{\mathcal{P}, \tau^{\star}_{\mathcal{P}, \mathcal{H}}}(h) = 0$.*

This definition ensures that there exists a hypothesis $h^{\star}$ in $\mathcal{H}$ that achieves an error rate below threshold $\tau^{\star}_{\mathcal{P}, \mathcal{H}}$ on every domain sampled from $\mathcal{P}$, and $\tau^{\star}_{\mathcal{P}, \mathcal{H}}$ represents the smallest such achievable threshold. Thus, $\tau^{\star}_{\mathcal{P}, \mathcal{H}}$ serves as a benchmark for our learning problem and any threshold $\tau \geq \tau^{\star}_{\mathcal{P}, \mathcal{H}}$ is achievable. We assume that $\tau^{\star}_{\mathcal{P}, \mathcal{H}}$ is reasonably small–if $\tau^{\star}_{\mathcal{P}, \mathcal{H}} = 0.5$, then no hypothesis in $\mathcal{H}$ can perform well across the domains–though we do not assume it is zero.

Then given a hypothesis class $\mathcal{H}$ and a threshold $\tau$, a learner $\mathcal{A}$ access to a set of i.i.d. sampled domains $G = \{\mathcal{D}_1, \mathcal{D}_2, \ldots, \mathcal{D}_n\}$, and i.i.d. data $S_i$ collected from each domain $\mathcal{D}_i$, with the goal of outputting a hypothesis $h := \mathcal{A}(\{S_1, \ldots, S_n\})$ that achieves error rate of at most $\tau$ under almost every domain. This can be formalized into the following learnability problem for domain generalization, where we focus on the domain sample complexity: the number $n$ of observed domains needed to achieve domain generalization.

**Definition 3.2** (($\tau, \alpha, \gamma, \delta$)-domain learnability). *For any $\tau, \alpha, \gamma, \delta \in (0, 1)$, we say $(\mathcal{H}, \mathcal{G})$ is $(\tau, \alpha, \gamma, \delta)$-learnable if there exists finite integers $n$ and $m$ for which there exists an algorithm $\mathcal{A}$ such that for any distribution $\mathcal{P}$ over $\mathcal{G}$ with optimal error bound $\tau^{\star}_{\mathcal{P}, \mathcal{H}} \leq \tau - \alpha$, with probability at least $1 - \delta$ over $G \sim \mathcal{P}^n$ and $S_i \sim \mathcal{D}_i^m$ for all $\mathcal{D}_i \in G$,*

$$\mathrm{Er}_{\mathcal{P}, \tau}(\mathcal{A}(\{S_1, \ldots, S_n\})) \leq \gamma.$$

*The domain sample complexity is the smallest integer $n$ satisfying the above constraint.*

# 4 Characterizing Domain Sample Complexity Using Domain Shattering Dimension

To characterize the domain sample complexity, we introduce the following combinatorial measure inspired by the fat-shattering dimension [Kearns and Schapire, 1994].

**Definition 4.1** (Domain shattering dimension). *Let $\mathcal{G}$ be a set of domains. We say a subset $S \subseteq \mathcal{G}$ is $\alpha$-shattered by $\mathcal{H}$ at $\tau$ if for all $E \subseteq S$, there exists a hypothesis $h_E \in \mathcal{H}$ satisfying*

$$\mathrm{err}_{\mathcal{D}}(h_E) < \tau - \alpha \qquad \text{for every } \mathcal{D} \in E,$$
$$\mathrm{err}_{\mathcal{D}}(h_E) > \tau \qquad \text{for every } \mathcal{D} \in S \setminus E.$$

*The domain shattering dimension of $\mathcal{H}$, denoted by $\mathrm{Gdim}(\mathcal{H}, \mathcal{G}, \tau, \alpha)$, is defined to be the maximum size of set $S$ that can be $\alpha$-shattered by $\mathcal{H}$ at $\tau$.*

**Monotonicity of domain shattering dimension.** It follows directly from the definition that for any $\tau, \alpha, \tau', \alpha'$, if $\tau' \geq \tau$ and $\tau' - \alpha' \leq \tau - \alpha$, we have $\mathrm{Gdim}(\mathcal{H}, \mathcal{G}, \tau', \alpha') \leq \mathrm{Gdim}(\mathcal{H}, \mathcal{G}, \tau, \alpha)$.

## 4.1 Upper Bound

We show an upper bound on the domain sample complexity using the domain shattering dimension in Theorem 4.1 below. We prove this upper bound using the following natural min-max variant of the empirical risk minimization (ERM) algorithm. In the next subsection (Section 4.2), we will prove a lower bound that matches our upper bound up to a polylogarithmic factor.

**Min-Max ERM Algorithm.** Given a set of i.i.d. sampled domains $G = \{\mathcal{D}_1, \mathcal{D}_2, \ldots, \mathcal{D}_n\}$, we assume access to approximate error rates for any hypothesis $h \in \mathcal{H}$ on each domain. Specifically, for some $\varepsilon > 0$, for every domain $\mathcal{D} \in G$ and hypothesis $h \in \mathcal{H}$, we can access an estimate $\widehat{\mathrm{err}}_{\mathcal{D}}(h)$ of the true error $\mathrm{err}_{\mathcal{D}}(h)$ such that

$$|\widehat{\mathrm{err}}_{\mathcal{D}}(h) - \mathrm{err}_{\mathcal{D}}(h)| < \varepsilon \quad \text{for all } \mathcal{D} \in G \text{ and } h \in \mathcal{H}. \tag{2}$$

By standard uniform convergence guarantees, with success probability at least $1 - \delta$, these estimates can be obtained as the empirical error on $O((\mathrm{VCdim}(\mathcal{H}) + \log(n/\delta))/\varepsilon^2)$ i.i.d. data points from each domain's distribution. Given access to $\widehat{\mathrm{err}}_{\mathcal{D}}(h)$, we return the min-max predictor

$$\widehat{h} = \arg\min_{h \in \mathcal{H}} \max_{\mathcal{D} \in G} \widehat{\mathrm{err}}_{\mathcal{D}}(h). \tag{3}$$

The following theorem upper bounds $\mathrm{Er}_{\mathcal{P}, \tau}(\widehat{h})$ in terms of the domain shattering dimension and the number of training domains. This implies a domain sample complexity upper bound.

**Theorem 4.1.** *Let $\mathcal{H}$ be a class of hypotheses $h : X \to \{0, 1\}$ and let $\mathcal{G}$ be a family of domains $\mathcal{D}$ each being a distribution over $X \times \{0, 1\}$. Define $d := \mathrm{Gdim}(\mathcal{H}, \mathcal{G}, \tau, \alpha)$ for $\tau, \alpha \in [0, 1]$. For every $\varepsilon, \delta \in (0, 1/2)$, for every domain distribution $\mathcal{P}$ over $\mathcal{G}$ satisfying $\tau^\star_{\mathcal{P}, \mathcal{H}} \leq \tau - \alpha - 2\varepsilon$, with probability at least $1 - \delta$ over a sample $G$ of $n$ i.i.d. domains drawn from $\mathcal{P}$, when given access to $\widehat{\mathrm{err}}_{\mathcal{D}}(h)$ for all $\mathcal{D} \in G$ and $h \in \mathcal{H}$ satisfying (2), the min-max predictor $\widehat{h}$ in (3) satisfies*

$$\mathrm{Er}_{\mathcal{P}, \tau}(\widehat{h}) \leq O\left(\frac{d \log^2 n + \log(1/\delta)}{n}\right).$$

Theorem 4.1 guarantees good accuracy of $\widehat{h}$ on new, unseen domains drawn from $\mathcal{P}$. To prove this result, we start by analyzing the performance of $\widehat{h}$ on the training domains in $G$. By the assumption of Theorem 4.1, there exists $h^\star \in \mathcal{H}$ such that $\mathrm{err}_{\mathcal{D}}(h^\star) \leq \tau - \alpha - 2\varepsilon$ for every $\mathcal{D} \in G$. Now by (2) and (3), we have

$$\mathrm{err}_{\mathcal{D}}(\widehat{h}) < \widehat{\mathrm{err}}_{\mathcal{D}}(\widehat{h}) + \varepsilon \leq \widehat{\mathrm{err}}_{\mathcal{D}}(h^\star) + \varepsilon < \mathrm{err}_{\mathcal{D}}(h^\star) + 2\varepsilon \leq \tau - \alpha \quad \text{for every } \mathcal{D} \in G. \tag{4}$$

This ensures that $\widehat{h}$ achieves low error on every training domain $\mathcal{D} \in G$. To prove Theorem 4.1, we need to show that $\widehat{h}$ achieves low error on new domains drawn from $\mathcal{P}$. Specifically, for some

$$\gamma = O\left(\frac{d \log^2 n + \log(1/\delta)}{n}\right),$$

our goal is to show that with probability at least $1 - \delta$, the $\widehat{h}$ returned by min-max ERM satisfies

$$\Pr_{\mathcal{D} \sim \mathcal{P}}[\mathrm{err}_{\mathcal{D}}(\widehat{h}) > \tau] \leq \gamma. \tag{5}$$

To prove the guarantee (5) for $\mathcal{D} \sim \mathcal{P}$ from the guarantee (4) for $\mathcal{D} \in G$, we establish a uniform convergence bound over all $h \in \mathcal{H}$.

One natural idea for establishing a desired uniform convergence bound is by applying existing results about the fat-shattering dimension Kearns and Schapire [1994], which is similar to our domain shattering dimension. However, this idea falls short for our purpose because 1) the fat-shattering dimension is defined as a maximum over *all* thresholds $\tau$, and 2) prior results from the fat-shattering dimension incur a constant-factor blow-up in the margin $\alpha$. We instead use results about partial concept classes from Alon et al. [2022]. Specifically, for each $h \in \mathcal{H}$, we construct a partial concept $f_h : \mathcal{G} \mapsto \{0, 1, \perp\}$ by letting

$$f_h(\mathcal{D}) = \begin{cases} 1 & \text{if } \mathrm{err}_{\mathcal{D}}(h) > \tau, \\ 0 & \text{if } \mathrm{err}_{\mathcal{D}}(h) < \tau - \alpha, \\ \perp & \text{o.w.} \end{cases}$$

This allows us to construct a new partial concept class $\mathcal{F} = \{f_h | h \in \mathcal{H}\}$. Our assumption in Theorem 4.1 that the domain shattering dimension of $\mathcal{H}$ is $d$ ensures that the VC dimension of $\mathcal{F}$ is $d$ (see Definitions 2.1 and 4.1). Now (4) can be equivalently written as

$$f_{\widehat{h}}(\mathcal{D}) = 0 \quad \text{for every } \mathcal{D} \in G,$$

and similarly, our goal (5) is equivalent to

$$\Pr_{\mathcal{D} \sim \mathcal{P}}[f_{\widehat{h}}(\mathcal{D}) = 1] \leq \gamma.$$

Thus, to prove Theorem 4.1, it suffices to establish the following uniform convergence bound:

$$\Pr_{G \sim \mathcal{P}^n}[\exists f \in \mathcal{F} \text{ s.t. } \Pr_{\mathcal{D} \sim \mathcal{P}}[f(\mathcal{D}) = 1] > \gamma \text{ and } \forall \mathcal{D} \in G, f(\mathcal{D}) = 0] \leq \delta.$$

We formally state this uniform convergence bound in the following general lemma:[4]

**Lemma 4.2** (Uniform convergence for partial concepts). *Let $\mathcal{F}$ be a class of partial concepts $f : Z \to \{0, 1, \perp\}$ on an arbitrary input space $Z$. Assume that $\mathcal{F}$ has VC dimension $d$. For every $n \in \mathbf{Z}_{>0}$ and $\delta \in (0, 1/2)$, there exists*

$$\gamma = O\left(\frac{d \log^2 n + \log(1/\delta)}{n}\right) \tag{6}$$

*such that for every distribution $\mathcal{P}$ over $Z$, for $n$ i.i.d. data points $z_1, \ldots, z_n$ drawn from $\mathcal{P}$,*

$$\Pr_{z_1, \ldots, z_n}[\exists f \in \mathcal{F} \text{ s.t. } \Pr_{z \sim \mathcal{P}}[f(z) = 1] > \gamma \text{ and } \forall i \in [n], f(z_i) = 0] \leq \delta. \tag{7}$$

We establish this uniform convergence bound using a standard symmetrization trick combined with a generalized Sauer-Shelah-Perles lemma for partial concept classes (Theorem B.1) by Alon et al. [2022]. We defer the full proof of Lemma 4.2 to Appendix B.1.

**Remark 4.1** (Choice of the threshold $\tau$). *It is important to note that while our upper bound depends on the choice of $\tau$ and $\alpha$, the algorithm itself does not. Let us fix the relationship $\tau_{\mathcal{P}, \mathcal{H}}^{\star} = \tau - \alpha - 2\varepsilon$. As we increase $\tau$ and $\alpha$ at the same rate–i.e., relax the target error threshold $\tau$–the value of $\mathrm{Gdim}(\mathcal{H}, \mathcal{G}, \tau, \alpha)$ decreases monotonically. This means that a larger proportion of domains can meet the relaxed threshold $\tau$. Therefore, the choice of $\tau$ captures a trade-off between the strictness of the generalization goal and the fraction of domains that are able to satisfy it.*

**Remark 4.2** (Beyond binary classification). *We emphasize that neither the algorithm nor the analysis relies on binary labels, and both can be directly applied to other learning tasks such as multi-class classification and regression.*

---

[4]We remark that a key message from the work of Alon et al. [2022] is that uniform convergence and the ERM algorithm both fail for learning partial concepts. This, however, does not contradict our Lemma 4.2. Roughly speaking, the uniform convergence needed in the setting of Alon et al. [2022] requires replacing $\Pr_{z \sim \mathcal{P}}[f(z) = 1]$ in (7) with $\Pr_{z \sim \mathcal{P}}[f(z) = 1 \text{ or } \perp]$. This stronger form of uniform convergence does not hold.

**Remark 4.3** (Standard ERM fails). *Note that standard ERM, which selects a hypothesis minimizing the empirical error over the entire pool of training data, may fail in our setting. Consider a toy example with two hypotheses: one incurs an error of $0.3$ on every domain, while the other achieves $0$ error on half of the domains and $0.5$ on the other half. Standard ERM would choose the latter, despite its poor worst-case performance across domains.*

## 4.2 Lower Bound

We prove a lower bound (Theorem 4.3) showing that the error upper bound in Theorem 4.1 is essentially information-theoretically optimal up to an $O(\log^2 n)$ factor. It would be ideal to show that the error bound is *instance-wise* optimal: for every fixed choice of $\mathcal{H}, \mathcal{G}$ and parameters $\tau, \alpha, \varepsilon, \delta$, the error bound in Theorem 4.1 cannot be improved even if one uses a learning algorithm specifically designed for those fixed choices. However, this perfect instance-wise optimality does not hold for the following trivial reason. For every domain $\mathcal{D} \in \mathcal{G}$, let $\mathcal{D}_X$ denote the marginal distribution of $x \in X$ where $(x, y) \sim \mathcal{D}$. Consider a fixed choice of $\mathcal{G}$ and suppose that the marginal distributions $\mathcal{D}_X$ for $\mathcal{D} \in \mathcal{G}$ are supported on disjoint subsets of $X$. In this case, the learner can simply always output the hypothesis $h^\star$ that simultaneously minimizes the error on every domain $\mathcal{D} \in \mathcal{G}$. This solves our domain generalization task without observing any training domains drawn from $\mathcal{P}$. The domain sample complexity is zero, but the domain shattering dimension can be very large, implying that our upper bound Theorem 4.1 is far from optimal in this setting.

We thus make a slight compromise in the level of instance-wise optimality. We still consider arbitrary fixed choices of $\mathcal{H}$ and $\mathcal{G}$. However, in our learning task, the domains do not solely come from $\mathcal{G}$, but instead come from a mildly extended family $\mathcal{G}'$. Importantly, for some domains $\mathcal{D}$ in the extended family $\mathcal{G}'$, their marginal distributions $\mathcal{D}_X$ will be supported on overlapping subsets of $X$. In this case, we are able to prove a domain sample complexity lower bound (Theorem 4.3) that nearly matches our upper bound (Theorem 4.1).

Concretely, we assume that there exists a distribution $\mathcal{D}_0$ of data points $(x, y) \in X \times \{0, 1\}$ such that $\mathrm{err}_{\mathcal{D}_0}(h) = 0$ for every $h \in \mathcal{H}$.[5] Let $d$ be the domain shattering dimension of $\mathcal{H}$ on $\mathcal{G}$, and let $\mathcal{D}_1, \ldots, \mathcal{D}_d \in \mathcal{G}$ be $d$ domains shattered by $\mathcal{H}$. For every $i = 1, \ldots, d$, we define a new domain $\mathcal{D}_i'$ as follows:

$$\mathcal{D}_i' = (1 - \lambda)\mathcal{D}_0 + \lambda(\neg \mathcal{D}_i). \tag{8}$$

Here, $\lambda \in [0, 1]$ is a fixed parameter, and $\neg \mathcal{D}_i$ is obtained by flipping the labels in $\mathcal{D}$. That is, $\neg \mathcal{D}_i$ is the distribution of $(x, \neg y)$ for $(x, y) \sim \mathcal{D}_i$. Equation (8) defines $\mathcal{D}_i'$ as a mixture of $\mathcal{D}_0$ and $\neg \mathcal{D}_i$.

We define $\mathcal{G}'$ to be $\mathcal{G} \cup \{\mathcal{D}_0, \mathcal{D}_1', \ldots, \mathcal{D}_d'\}$. We view $\mathcal{G}'$ as a mild extension of $\mathcal{G}$. In particular, as we show in Theorem 4.3 below, $\mathcal{H}$ has the same domain shattering dimension on $\mathcal{G}'$ as on $\mathcal{G}$ for an appropriate choice of $\lambda$. Moreover, in Theorem 4.3 we show a domain sample complexity lower bound on $\mathcal{G}'$ that nearly matches our upper bound in Theorem 4.1.

**Theorem 4.3** (Lower bound). *Consider an arbitrary class $\mathcal{H}$ of hypotheses $h : X \to \{0, 1\}$ and an arbitrary family $\mathcal{G}$ of domains $\mathcal{D}$ (i.e. distributions over $X \times \{0, 1\}$). Assume that $\mathrm{Gdim}(\mathcal{H}, \mathcal{G}, \tau, \alpha) = d$ for some $\tau, \alpha \in \mathbb{R}$ and $d \in \mathbf{Z}_{>0}$, where $0 \leq \alpha < \tau \leq 1/2$. Let $\mathcal{D}_0$ be a distribution of $(x, y) \in X \times \{0, 1\}$ satisfying $\mathrm{err}_{\mathcal{D}_0}(h) = 0$ for every $h \in \mathcal{H}$. Let $\mathcal{D}_1, \ldots, \mathcal{D}_d \in \mathcal{G}$ be $d$ domains shattered by $\mathcal{H}$, and define $\mathcal{G}' = \mathcal{G} \cup \{\mathcal{D}_0, \mathcal{D}_1', \ldots, \mathcal{D}_d'\}$ as in (8) for*

$$\lambda = \frac{\tau - \alpha}{1 - \tau} \in (0, 1]. \tag{9}$$

*Then $\mathrm{Gdim}(\mathcal{H}, \mathcal{G}', \tau, \alpha) = d$. Moreover, for some $\gamma > 0, \delta \in (0, 1/4), n \in \mathbf{Z}_{>0}$, and an error threshold $\tau' < \tau - \frac{1-\tau}{1-\alpha} \cdot \alpha \in (\tau - \alpha, \tau]$, suppose there is an algorithm $A$ with the following property on every distribution $\mathcal{P}$ over $\mathcal{G}'$ satisfying $\tau_{\mathcal{P},\mathcal{H}}^\star < \tau - \alpha$: it takes $n$ domains drawn i.i.d. from $\mathcal{P}$ as input, and with probability at least $1 - \delta$, outputs a hypothesis $\widehat{h}$ such that*

$$\mathrm{Er}_{\mathcal{P},\tau'}(\widehat{h}) \leq \gamma.$$

*Then*

$$\gamma = \Omega\left(\min\left\{1, \frac{d + \log(1/\delta)}{n}\right\}\right). \tag{10}$$

---

[5]We view this as a mild assumption. Note that $\mathcal{D}_0$ does not need to be a member of $\mathcal{G}$. This assumption is satisfied as long as there is an input point $x_0 \in X$ receiving the same label $h(x_0) = b \in \{0, 1\}$ for all $h \in \mathcal{H}$, in which case we choose $\mathcal{D}_0$ as the degenerate distribution supported on the single point $(x_0, b)$.

We remark that the lower bound in Theorem 4.3 is for a lower (i.e., more challenging) error threshold $\tau'$ instead of the original error threshold $\tau$ in the definition of the domain shattering dimension. Nonetheless, the lower threshold $\tau'$ is always allowed to be above the optimal error $\tau^\star_{\mathcal{P},\mathcal{H}}$ (i.e. $\tau' > \tau - \alpha > \tau^\star_{\mathcal{P},\mathcal{H}}$).

We defer the proof of Theorem 4.3 to Appendix B.2.

# 5 Relationship between Domain Shattering Dimension and VC Dimension

In this section, we study the relationship between the VC dimension and the domain shattering dimension of a hypothesis class $\mathcal{H}$. It is easy to see that the domain shattering dimension can be much smaller than the VC dimension when the domains in $\mathcal{G}$ are similar—in the extreme case where $\mathcal{G}$ contains only a single domain, the domain shattering dimension cannot be more than one. We thus focus on the other direction: can the domain shattering dimension of a hypothesis class exceed its VC dimension? How much larger can it be?

We give an accurate answer to this question in Theorems 5.1 and 5.2 below: for an error margin $\alpha$ (see Definition 4.1), the domain shattering dimension can be as large as $\Omega(d \log(1/\alpha))$, which is also the largest possible (up to a constant factor). Therefore, the domain shattering dimension can be arbitrarily larger than the VC dimension as $\alpha$ approaches zero, but for any fixed $\alpha > 0$, the domain shattering dimension is upper bounded linearly by the VC dimension. This implies that a PAC learnable hypothesis class is also learnable for our domain generalization task in Definition 3.2.

**Theorem 5.1.** *For every positive integer $d$, there exists a hypothesis class $\mathcal{H}$ with $\mathrm{VCdim}(\mathcal{H}) = d$ satisfying the following property. For any $\alpha \in (0, 1/12)$, there exist $k = \Omega(d \log(1/\alpha))$ domains $\mathcal{D}_1, \ldots, \mathcal{D}_k$ such that*
$$\mathrm{Gdim}(\mathcal{H}, \{\mathcal{D}_1, \ldots, \mathcal{D}_k\}, 0.3, \alpha) = k.$$

**Theorem 5.2.** *Let $\mathcal{H}$ be an arbitrary hypothesis class with VC dimension $d$. For any set $\mathcal{G}$ of domains, any threshold $\tau \in \mathbb{R}$, and any margin $\alpha \in (0, 1/2)$,*
$$\mathrm{Gdim}(\mathcal{H}, \mathcal{G}, \tau, \alpha) = O(d \log(1/\alpha)). \tag{11}$$

We defer the proofs of these two theorems to Appendices B.3 and B.4. Our proof of Theorem 5.2 uses a dimension reduction argument combined with the standard Sauer-Shelah-Perles lemma, which is inspired by the proof of a classic covering number upper bound in terms of the VC dimension [see e.g. Vershynin, 2018, Theorem 8.3.18].

# 6 Connection to Domain Adaptation

In the domain adaptation literature, a common setting assumes that the training data are drawn from a source distribution, while the test data come from a different but related target distribution. When the source and target distributions are sufficiently similar, a model trained on the source data can generalize well to the target distribution. To quantify this similarity, various notions have been proposed, such as the $\mathcal{H}$-divergence [Ben-David et al., 2010] and the propensity scoring function class [Kim et al., 2022].

Our work makes no assumptions about the inter-domain structure, except that the optimal domain error bound $\tau^\star_{\mathcal{P},\mathcal{H}}$ is reasonably small. We then upper bound the domain generalization error using the domain shattering dimension. In this section, we show that when the covering number of the domain space $\mathcal{G}$ w.r.t. the $\mathcal{H}$-divergence is small, the domain shattering dimension is also small. In fact, we show a stronger result (Theorem 6.1) for a refined variant of the $\mathcal{H}$-divergence. This illustrates that our domain shattering dimension can capture the similarity between domains without requiring an explicit model of their relationships.

Inspired by the $\mathcal{H}$-divergence introduced by Ben-David et al. [2010], where
$$d_{\mathcal{H}}(\mathcal{D}, \mathcal{D}') := \sup_{h \in \mathcal{H}} |\mathrm{err}_{\mathcal{D}}(h) - \mathrm{err}_{\mathcal{D}'}(h)|,$$

we define a refined notion of divergence. Given $\mathcal{H}, \tau$, the $(\mathcal{H}, \tau)$-divergence between two domains $\mathcal{D}, \mathcal{D}'$ is
$$d_{\mathcal{H},\tau}(\mathcal{D}, \mathcal{D}') := \sup_{h \in \mathcal{H}:\min\{\mathrm{err}_{\mathcal{D}}(h), \mathrm{err}_{\mathcal{D}'}(h)\} \leq \tau} |\mathrm{err}_{\mathcal{D}}(h) - \mathrm{err}_{\mathcal{D}'}(h)|.$$

Note that $d_{\mathcal{H},\tau}(\mathcal{D}, \mathcal{D}') \leq d_{\mathcal{H}}(\mathcal{D}, \mathcal{D}')$ as we only take supremum over a subset of hypotheses $\{h \in \mathcal{H} : \min\{\text{err}_{\mathcal{D}}(h), \text{err}_{\mathcal{D}'}(h)\} \leq \tau\}$. Given domain space $\mathcal{G}$, we say $\mathcal{G}'$ is an $\alpha$-cover of $\mathcal{G}$ w.r.t. $d_{\mathcal{H},\tau}$ if for every $\mathcal{D} \in \mathcal{G}$, there exists $\mathcal{D}' \in \mathcal{G}'$ such that $d_{\mathcal{H},\tau}(\mathcal{D}, \mathcal{D}') \leq \alpha$. Any $\alpha$-cover $\mathcal{G}'$ w.r.t. $d_{\mathcal{H}}$ is also an $\alpha$-cover w.r.t. $d_{\mathcal{H},\tau}$.

**Theorem 6.1.** *For every hypothesis class $\mathcal{H}$, domain space $\mathcal{G}$ and $\tau, \alpha \in (0,1)$, let $\mathcal{G}'$ be an $\frac{\alpha}{2}$-cover of $\mathcal{G}$ w.r.t. $d_{\mathcal{H},\tau}$. We have $\text{Gdim}(\mathcal{H}, \mathcal{G}, \tau, \alpha) \leq |\mathcal{G}'|$.*

*Proof of Theorem 6.1.* For every $\mathcal{D}' \in \mathcal{G}'$, define $\mathcal{G}(\mathcal{D}') := \{\mathcal{D} \in \mathcal{G} : d_{\mathcal{H},\tau}(\mathcal{D}, \mathcal{D}') \leq \frac{\alpha}{2}\}$. Since $\mathcal{G}'$ is an $\frac{\alpha}{2}$-cover, the union of $\mathcal{G}(\mathcal{D}')$ over $\mathcal{D}' \in \mathcal{G}'$ is $\mathcal{G}$.

Fix an arbitrary $\mathcal{D}' \in \mathcal{G}'$ and let $\mathcal{D}_1, \mathcal{D}_2$ be two domains in $\mathcal{G}(\mathcal{D}')$. We show that $\mathcal{D}_1, \mathcal{D}_2$ cannot be $\alpha$-shattered at $\tau$. In particular, we show that for every $h \in \mathcal{H}$, the two conditions $\text{err}_{\mathcal{D}_1}(h) < \tau - \alpha$ and $\text{err}_{\mathcal{D}_2}(h) > \tau$ cannot hold simultaneously. Indeed, if $\text{err}_{\mathcal{D}_1}(h) < \tau - \alpha$, then we have

$$\text{err}_{\mathcal{D}'}(h) \leq \text{err}_{\mathcal{D}_1}(h) + \frac{\alpha}{2} < \tau - \frac{\alpha}{2},$$

and consequently,

$$\text{err}_{\mathcal{D}_2}(h) \leq \text{err}_{\mathcal{D}'}(h) + \frac{\alpha}{2} < \tau.$$

Therefore, $\text{err}_{\mathcal{D}_2}(h) > \tau$ cannot hold.

We have now shown that every subset of domains that is $\alpha$-shattered at $\tau$ contains at most one domain from each $\mathcal{G}(\mathcal{D}')$ for $\mathcal{D}' \in \mathcal{G}'$. Thus the domain shattering dimension $\text{Gdim}(\mathcal{H}, \mathcal{G}, \tau, \alpha)$ does not exceed $|\mathcal{G}'|$. $\qquad\square$

When all the domains are similar, the cover size is 1, then sampling one domain is sufficient. Below are two examples of domain spaces with cover size 1.

**Example 6.1** (Small $\mathcal{H}$-divergences)**.** *For any domain space $\mathcal{G}$, if for every pair of distributions $\mathcal{D}, \mathcal{D}' \in \mathcal{G}$, $d_{\mathcal{H}}(\mathcal{D}, \mathcal{D}') \leq \alpha$. Then the $\alpha$-cover size of $\mathcal{G}$ is 1.*

**Example 6.2** (Smooth Distributions)**.** *Given a marginal data distribution $\mu$ and a labeling function $p^\star$, we define $\mathcal{D}_{\mu,p^\star}$ as the distribution over labeled data where features are drawn from $\mu$ and labels are assigned according to $p^\star$. For a parameter $\gamma \in (0,1)$, a labeling function $p^\star$, and a base marginal distribution $\mu_0$, define the smooth domain space as*

$$\mathcal{G}_{p^\star,\mu_0,\gamma} = \left\{ \mathcal{D}_{\mu,p^\star} \;\middle|\; \frac{\mu(x)}{\mu_0(x)} \in [\gamma, \tfrac{1}{\gamma}], \; \forall x \in supp(\mu_0) \right\}.$$

*For domain space $\mathcal{G}_{p^\star,\mu_0,\gamma}$, we can find a $\left(\frac{1}{\gamma^2} - 1\right) \tau$-cover of size 1.*

For any two distributions $\mathcal{D}_{\mu_1,p^\star}, \mathcal{D}_{\mu_2,p^\star} \in \mathcal{G}_{p^\star,\mu_0,\gamma}$, it holds that $\frac{\mu_1(x)}{\mu_2(x)} \in [\gamma^2, \frac{1}{\gamma^2}]$ for all $x \in supp(\mu_0)$. Consequently, for any hypothesis $h$, we have

$$\gamma^2 \cdot \text{err}_{\mathcal{D}_{\mu_2,p^\star}}(h) \leq \text{err}_{\mathcal{D}_{\mu_1,p^\star}}(h) \leq \frac{1}{\gamma^2} \cdot \text{err}_{\mathcal{D}_{\mu_2,p^\star}}(h).$$

Therefore, by the definition of the $(\mathcal{H}, \tau)$-divergence, it follows that

$$d_{\mathcal{H},\tau}(\mathcal{D}_{\mu_1,p^\star}, \mathcal{D}_{\mu_2,p^\star}) \leq \left(\frac{1}{\gamma^2} - 1\right)\tau.$$

Hence, for domain space $\mathcal{G}_{p^\star,\mu_0,\gamma}$, we can find a $\left(\frac{1}{\gamma^2} - 1\right)\tau$-cover of size 1.

## 7 Discussion and Limitations

This work was inspired by the very real and widely acknowledged problem of transferring research from well-funded flagship medical research institutions to essentially all communities, even those with no direct access to these facilities, "from preeminent bench to geographically remote bedside"

(Deng et al. [2020]). Our succinct characterization and algorithm close some facets of this central question, while opening others.

**How to compute the shattering dimension.** As discussed in Remark 4.1, the choice of $\tau$ reflects a trade-off between the strictness of the generalization goal and the fraction of domains that can satisfy it, as captured by $\mathrm{Gdim}(\mathcal{H}, \mathcal{G}, \tau, \alpha)$. However, choosing an appropriate $\tau$ depends on understanding the value of $\mathrm{Gdim}(\mathcal{H}, \mathcal{G}, \tau, \alpha)$, which–like the VC dimension–is generally difficult to compute. Developing tools or approximation techniques to estimate $\mathrm{Gdim}$ is an important direction for future work.

**Choice of hypothesis class $\mathcal{H}$.** We focus on the setting where a hypothesis class $\mathcal{H}$ is given and assumed to contain a reasonably good hypothesis that generalizes across all domains. However, in practice, one may have access to a collection of candidate classes. How should we select the best class–one that contains a reasonably good hypothesis but is not overly complex–for generalization? This question relates to structured ERM in the learning theory literature and presents an interesting open direction.

**Gap between upper and lower bounds.** Although we show that the domain shattering dimension provides both upper and lower bounds on the domain sample complexity, a small gap remains. Our upper bound guarantees that $O(\mathrm{Gdim}(\mathcal{H}, \mathcal{G}, \tau, \alpha))$ sampled domains are sufficient to learn a predictor with error at most $\tau$ on most domains, while the lower bound shows that $\Omega(\mathrm{Gdim}(\mathcal{H}, \mathcal{G}, \tau, \alpha))$ domains are necessary to achieve an error threshold $\tau' \in (\tau - \alpha, \tau)$. Additionally, the lower bound does not apply to all domain families $\mathcal{G}$; in some cases, the construction requires augmenting $\mathcal{G}$ with a few extra domains. It would be of moderate interest to close this gap.

**Unlabeled data from some unseen groups.** In this work, we assume no information is available from unobserved domains. However, in real-world applications such as healthcare, labeled data can be expensive—particularly in rural regions where diagnostic tools may be limited—while unlabeled data are often cheaper and more accessible. This brings us full circle to the motivating scenario in the original paper on domain generalization Blanchard et al. [2011], what is there termed automatic gating of flow cytometry data. In this problem, each patient yields a patient-specific distribution (i.e. domain) of $d$-vectors of attributes of cells. The label might capture whether or not the given cell is a lymphocyte. Our work strengthens the results of Blanchard et al. [2011] by obtaining generalization to all unseen patients, and not just in expectation over unseen patients. This inspiring scenario, in which unlabled data are plentiful, raises the following question: If we permit the (fixed) learned hypothesis to be modified based on unlabeled samples from the test domain, can we beat our lower bound and get away with training on *fewer* than $\mathrm{Gdim}(\mathcal{H}, \mathcal{G}, \tau, \alpha)$ domains, while maintaining the learning objective of performing well on *every* unseen domain?

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

# A  Deferred Related Work

Many previous theoretical works on domain generalization make explicit assumptions on how domains are related. Blanchard et al. [2011] formulated the problem and approached it using kernel methods. They and Muandet et al. [2013] differ from our work in assuming similarity among the various domains (possibly after a learned transformation); moreover, they achieve low error only in expectation over the choice of test domain. More recently, Mohri et al. [2019] assumes that the learner has access to a family of training domains, and the test domains are a family of mixtures (i.e., convex combinations) of the training domains. The coefficients of the convex combinations are assumed to be in a restricted range. The works of Shao et al. [2022], Montasser et al. [2024] assume that the domains are related by transformations. These works [Mohri et al., 2019, Shao et al., 2022, Montasser et al., 2024] all use a min-max objective similar to our work. Ben-David et al. [2010] studies learning with a source (training) domain and a target (test) domain with the assumption that the each hypothesis $h \in \mathcal{H}$ has similar overall average on the two domains (they define $\mathcal{H}$-divergence as the largest gap between the two averages). Kim et al. [2022] assume that the ratios of the probability density between domains (i.e. propensity scores) are bounded and come from a restricted family. Schölkopf et al. [2012], Zhang et al. [2015], Gong et al. [2016] formulate the problem from a causal perspective and make corresponding causal assumptions.

Inspired by causal reasoning, and seeking to eliminate spurious correlations, Arjovsky et al. [2019] introduced *Invariant Risk Minimization*, in which the different domains correspond to training data collected in different environments which "could represent different measuring circumstances locations, times, experimental conditions" and so on. Invariant Risk Minimization seeks a representation mapping for instances, under which the optimal classifier is identical for all environments. Deng et al. [2020], whose setting and assumptions exactly match ours, also seeks a representation approach, specifically, via adversarial learning using techniques developed in Madras et al. [2018]; their algorithmic results have provable guarantees for unseen domains only when the observed domains form a cover with respect to $\mathcal{H}$-divergence for the set of possible domains.

The work of Garg et al. [2021] studies domain generalization with the goal of minimizing the average error across domains, which is equivalent to the error on the single large domain defined as the mixture of all the domains in the problem. This makes their learning task similar to the standard PAC learning with the additional advantage that the training data is grouped by domain. This advantage allows them to show computationally efficient learning algorithms for specific learning problems that do not have known efficient algorithms in the standard PAC learning setting.

Alon et al. [2024] study a related meta-learning problem, where instead of outputting a single hypothesis $h$, their learner outputs a hypothesis class which can be used later to learn a good hypothesis on a new task (analogous to a new domain) given additional data from that task. They focus on the realizable setting with performance measured using the average accuracy over random new tasks. A large body of work studies meta-learning for linear regression and related problems under the assumption that the underlying parameters of different tasks can be represented in a shared low-dimensional subspace [Maurer et al., 2016, Kong et al., 2020, Tripuraneni et al., 2020, 2021, Du et al., 2021, Collins et al., 2021, Thekumparampil et al., 2021, Duchi et al., 2022, Aliakbarpour et al., 2024].

There is a large body of recent and earlier work on multi-distribution learning [Larsen et al., 2024, Zhang et al., 2024, Peng, 2024, Awasthi et al., 2023, Haghtalab et al., 2022, Chen et al., 2018, Nguyen and Zakynthinou, 2018, Blum et al., 2017]. This problem is closely related to ours in that they also do not impose structural assumptions on the domains, and they also consider the min-max objective. However, their goal is to learn a hypothesis that performs well on the *observed* domains, so their focus lies in the total number of data points required for learning (i.e., the sample or query complexity). In contrast, we aim to generalize to *unobserved* domains and focus on the number of *domains* the learner needs to observe–the domain sample complexity.

# B  Deferred Proofs

## B.1  Proof of Lemma 4.2

**Lemma 4.2** (Uniform convergence for partial concepts). *Let $\mathcal{F}$ be a class of partial concepts $f : Z \to \{0, 1, \bot\}$ on an arbitrary input space $Z$. Assume that $\mathcal{F}$ has VC dimension $d$. For every*

$n \in \mathbf{Z}_{>0}$ and $\delta \in (0, 1/2)$, there exists

$$\gamma = O\left(\frac{d \log^2 n + \log(1/\delta)}{n}\right) \tag{6}$$

such that for every distribution $\mathcal{P}$ over $Z$, for $n$ i.i.d. data points $z_1, \ldots, z_n$ drawn from $\mathcal{P}$,

$$\Pr_{z_1, \ldots, z_n} [\exists f \in \mathcal{F} \text{ s.t. } \Pr_{z \sim \mathcal{P}}[f(z) = 1] > \gamma \text{ and } \forall i \in [n], f(z_i) = 0] \leq \delta. \tag{7}$$

We will use the following result.

**Theorem B.1** (Quasipolynomial Sauer-Shelah-Perles Lemma for Disambiguations of Partial Concepts [Alon et al., 2022]). *Let $\mathcal{F}$ be a class of partial concepts $f : Z \to \{0, 1, \bot\}$ defined on an arbitrary domain $Z$ with VC dimension $d$. Let $S$ be a subset of $Z$ with size $|S| = n > 1$. Then there exists a class $\overline{\mathcal{F}}$ of total concepts $\overline{f} : S \to \{0, 1\}$ with size*

$$|\overline{\mathcal{F}}| = n^{O(d \log n)}$$

*that satisfies the following. For every $f \in \mathcal{F}$, there exists $\overline{f} \in \overline{\mathcal{F}}$ such that for every $s \in S$ with $f(s) \in \{0, 1\}$, it holds that $\overline{f}(s) = f(s)$.*

*Proof of Lemma 4.2.* We apply the symmetrization trick. Specifically, we independently draw another $n$ data points $z_1', \ldots, z_n'$ i.i.d. from $\mathcal{P}$. By the multiplicative Chernoff bound, assuming $n > C/\gamma$ for a sufficiently large absolute constant $C > 0$ (which can be guaranteed by the choice of $\gamma$ in (6)), for a fixed $f \in \mathcal{F}$ satisfying $\Pr_{z \sim \mathcal{P}}[f(z) = 1] > \gamma$, with probability at least $1/2$ we have

$$|\{i \in [n] : f(z_i') = 1\}| > \gamma n/2.$$

Therefore, to show (7), it suffices to show that

$$\Pr_{z_1, \ldots, z_n; z_1', \ldots, z_n'} [\exists f \in \mathcal{F} \text{ s.t. } |\{i \in [n] : f(z_i') = 1\}| > \gamma n/2 \text{ and } \forall i \in [n], f(z_i) = 0] \leq \delta/2. \tag{12}$$

The $2n$ data points $z_1, \ldots, z_n, z_1', \ldots, z_n'$ can be sampled equivalently as follows. We first draw $2n$ i.i.d. data points $r_1, \ldots, r_{2n}$ from $\mathcal{P}$, choose $n$ data points from them uniformly at random without replacement as $z_1, \ldots, z_n$, and define the remaining $n$ data points as $z_1', \ldots, z_n'$. We define $\mathbf{r} := (r_1, \ldots, r_{2n}), \mathbf{z} := (z_1, \ldots, z_n), \mathbf{z}' := (z_1', \ldots, z_n')$.

For a fixed $\mathbf{r}$, by Theorem B.1, there exists a class $\overline{\mathcal{F}}$ of total concepts $\overline{f} : \{r_1, \ldots, r_{2n}\} \to \{0, 1\}$ with

$$|\overline{\mathcal{F}}| \leq (2n)^{O(d \log(2n))} \tag{13}$$

satisfying the following property: for every $f \in \mathcal{F}$, there exists $\overline{f} \in \overline{\mathcal{F}}$ such that for every $i \in [2n]$ with $f(r_i) \in \{0, 1\}$, it holds that $\overline{f}(r_i) = f(r_i)$. Therefore, to show (12), it suffices to show that for every fixed choice of $\mathbf{r}$, we can bound the following conditional probability given $\mathbf{r}$:

$$\Pr_{\mathbf{z}, \mathbf{z}'}[\exists \overline{f} \in \overline{\mathcal{F}} \text{ s.t. } |\{i \in [n] : \overline{f}(z_i') = 1\}| > \gamma n/2 \text{ and } \forall i \in [n], \overline{f}(z_i) = 0 \mid \mathbf{r}] \leq \delta/2,$$

where the probability is over the random partitioning of $\mathbf{r}$ into $\mathbf{z}$ and $\mathbf{z}'$. By the union bound, it suffices to show that for every $\overline{f} \in \overline{\mathcal{F}}$,

$$\Pr_{\mathbf{z}, \mathbf{z}'}[|\{i \in [n] : \overline{f}(z_i') = 1\}| > \gamma n/2 \text{ and } \forall i \in [n], \overline{f}(z_i) = 0 \mid \mathbf{r}] \leq \delta/(2|\overline{\mathcal{F}}|).$$

Note that $|\{i \in [n] : \overline{f}(z_i') = 1\}| > \gamma n/2$ implies that among the $2n$ data points $r_1, \ldots, r_{2n}$, more than $\gamma/4$ fraction of the data points $r_i$ satisfy $\overline{f}(r_i) = 1$. Conditioned on $\mathbf{r}$ satisfying this property, since $z_1, \ldots, z_n$ are chosen randomly without replacement from $r_1, \ldots, r_{2n}$, the probability that all the $n$ data points $z_1, \ldots, z_n$ satisfy $\overline{f}(z_i) = 0$ is at most $(1 - \gamma/4)^n$. Therefore, it suffices to prove that

$$(1 - \gamma/4)^n \leq \delta/(2|\overline{\mathcal{F}}|).$$

This holds when

$$\gamma \geq \frac{C}{n} \log(|\overline{\mathcal{F}}|/\delta)$$

for a sufficiently large absolute constant $C > 0$. By (13), the above inequality can be achieved by a choice of $\gamma$ satisfying (6). $\square$

## B.2 Proof of Theorem 4.3

**Theorem 4.3** (Lower bound). *Consider an arbitrary class $\mathcal{H}$ of hypotheses $h : X \to \{0,1\}$ and an arbitrary family $\mathcal{G}$ of domains $\mathcal{D}$ (i.e. distributions over $X \times \{0,1\}$). Assume that $\mathrm{Gdim}(\mathcal{H}, \mathcal{G}, \tau, \alpha) = d$ for some $\tau, \alpha \in \mathbb{R}$ and $d \in \mathbf{Z}_{>0}$, where $0 \le \alpha < \tau \le 1/2$. Let $\mathcal{D}_0$ be a distribution of $(x, y) \in X \times \{0,1\}$ satisfying $\mathrm{err}_{\mathcal{D}_0}(h) = 0$ for every $h \in \mathcal{H}$. Let $\mathcal{D}_1, \ldots, \mathcal{D}_d \in \mathcal{G}$ be $d$ domains shattered by $\mathcal{H}$, and define $\mathcal{G}' = \mathcal{G} \cup \{\mathcal{D}_0, \mathcal{D}'_1, \ldots, \mathcal{D}'_d\}$ as in (8) for*

$$\lambda = \frac{\tau - \alpha}{1 - \tau} \in (0, 1]. \tag{9}$$

*Then $\mathrm{Gdim}(\mathcal{H}, \mathcal{G}', \tau, \alpha) = d$. Moreover, for some $\gamma > 0, \delta \in (0, 1/4), n \in \mathbf{Z}_{>0}$, and an error threshold $\tau' < \tau - \frac{1-\tau}{1-\alpha} \cdot \alpha \in (\tau - \alpha, \tau]$, suppose there is an algorithm $A$ with the following property on every distribution $\mathcal{P}$ over $\mathcal{G}'$ satisfying $\tau^\star_{\mathcal{P}, \mathcal{H}} < \tau - \alpha$: it takes $n$ domains drawn i.i.d. from $\mathcal{P}$ as input, and with probability at least $1 - \delta$, outputs a hypothesis $\widehat{h}$ such that*

$$\mathrm{Er}_{\mathcal{P}, \tau'}(\widehat{h}) \le \gamma.$$

*Then*

$$\gamma = \Omega\left(\min\left\{1, \frac{d + \log(1/\delta)}{n}\right\}\right). \tag{10}$$

**Claim B.2.** *In the setting of Theorem 4.3, for every $i = 1, \ldots, d$, and every hypothesis $h : X \to \{0,1\}$, we have*

$$\mathrm{err}_{\mathcal{D}'_i}(h) \ge \lambda\left(1 - \mathrm{err}_{\mathcal{D}_i}(h)\right), \tag{14}$$

$$\max\{\mathrm{err}_{\mathcal{D}'_i}(h), \mathrm{err}_{\mathcal{D}_i}(h)\} \ge \frac{\lambda}{1 + \lambda} = \tau - \frac{1 - \tau}{1 - \alpha} \cdot \alpha > \tau'. \tag{15}$$

*Moreover, the inequality (14) becomes an equality when $h \in \mathcal{H}$.*

*Proof.* By (8), for any hypothesis $h$, we have

$$\mathrm{err}_{\mathcal{D}'_i}(h) = (1 - \lambda)\mathrm{err}_{\mathcal{D}_0}(h) + \lambda(1 - \mathrm{err}_{\mathcal{D}_i}(h)).$$

Inequality (14) follows immediately from the trivial fact that $\mathrm{err}_{\mathcal{D}_0}(h) \ge 0$. When $h \in \mathcal{H}$, we have $\mathrm{err}_{\mathcal{D}_0}(h) = 0$, so (14) becomes an equality.

It remains to prove (15). Suppose $\mathrm{err}_{\mathcal{D}_i}(h) < \frac{\lambda}{1+\lambda}$. By (14),

$$\mathrm{err}_{\mathcal{D}'_i}(h) \ge \lambda\left(1 - \frac{\lambda}{1 + \lambda}\right) = \frac{\lambda}{1 + \lambda}.$$

This proves the first inequality in (15). The equality in (15) follows from the definition of $\lambda$ in (9). The last inequality in (15) is our assumption about $\tau'$ in Theorem 4.3. $\square$

*Proof of Theorem 4.3.* We first prove that $\mathrm{Gdim}(\mathcal{H}, \mathcal{G}', \tau, \alpha) = d$. By Claim B.2, for any $\mathcal{D}_i$ and the corresponding $\mathcal{D}'_i$ in (8), the errors of any hypothesis $h \in \mathcal{H}$ on $\mathcal{D}_i$ and $\mathcal{D}'_i$ are related as follows:

$$\mathrm{err}_{\mathcal{D}_i}(h) = 1 - \mathrm{err}_{\mathcal{D}'_i}(h)/\lambda.$$

Therefore,

$$\mathrm{err}_{\mathcal{D}'_i}(h) < \tau - \alpha \implies \mathrm{err}_{\mathcal{D}_i}(h) > 1 - (\tau - \alpha)/\lambda = \tau,$$
$$\mathrm{err}_{\mathcal{D}'_i}(h) > \tau \implies \mathrm{err}_{\mathcal{D}_i}(h) < 1 - \tau/\lambda \le 1 - (\tau - \alpha)/\lambda - \alpha = \tau - \alpha, \tag{16}$$

where we used the definition of $\lambda$ in (9). Consider any set $S \subseteq \mathcal{G}'$ shattered by $\mathcal{H}$. Clearly $\mathcal{D}_0$ cannot appear in $S$ because $\mathrm{err}_{\mathcal{D}_0}(h) = 0$ for every $h \in \mathcal{H}$. By (16), $\mathcal{D}_i$ and $\mathcal{D}'_i$ cannot both belong to $S$, and replacing $\mathcal{D}'_i$ with $\mathcal{D}_i$ in $S$ still leads to a shattered set. We can thus change $S$ into a shattered subset of $\mathcal{G}$ without changing its size. This implies that the domain shattering dimension of $\mathcal{H}$ on $\mathcal{G}'$ does not exceed its domain shattering dimension on $\mathcal{G}$. The reverse direction is trivial, so the two domain shattering dimensions are both equal to $d$.

Now we prove the lower bound (10) on $\gamma$. We can assume without loss of generality that $\gamma < 1/8$, because otherwise we already have $\gamma = \Omega(1)$ as needed to establish the theorem. For every bit string $b \in \{0,1\}^d$, we define domains $\overline{\mathcal{D}}_0, \ldots, \overline{\mathcal{D}}_d$ as follows:

$$\overline{\mathcal{D}}_0 = \mathcal{D}_0,$$

$$\overline{\mathcal{D}}_i = \begin{cases} \mathcal{D}_i, & \text{if } b_i = 0, \\ \mathcal{D}'_i, & \text{if } b_i = 1, \end{cases} \quad \text{for } i = 1, \ldots, d. \tag{17}$$

We define $\mathcal{P}_b$ to be the distribution that puts $1 - 4\gamma$ probability mass on $\overline{\mathcal{D}}_0$, and distributes the remaining $4\gamma$ probability mass uniformly on $\overline{\mathcal{D}}_1, \ldots, \overline{\mathcal{D}}_d$.

We first show that $\tau^\star_{\mathcal{P}_b, \mathcal{H}} < \tau - \alpha$. Since $\mathcal{D}_1, \ldots, \mathcal{D}_d$ are shattered by $\mathcal{H}$, there exists a hypothsis $h_b \in \mathcal{H}$ such that for every $i = 1, \ldots, d$,

$$\begin{aligned} \mathrm{err}_{\mathcal{D}_i}(h_b) &< \tau - \alpha, & \text{if } b_i = 0, \\ \mathrm{err}_{\mathcal{D}_i}(h_b) &> \tau, & \text{if } b_i = 1. \end{aligned}$$

When $\mathrm{err}_{\mathcal{D}_i}(h_b) > \tau$, by Claim B.2,

$$\mathrm{err}_{\mathcal{D}'_i}(h_b) < \lambda(1 - \tau) = \frac{\tau - \alpha}{1 - \tau} \cdot (1 - \tau) = \tau - \alpha.$$

Therefore, $\mathrm{err}_{\overline{\mathcal{D}}_i}(h_b) < \tau - \alpha$ for every $i = 0, 1, \ldots, d$. This completes the proof of our claim that $\tau^\star_{\mathcal{P}_b, \mathcal{H}} < \tau - \alpha$.

Now let us consider the following process. We first draw $b$ uniformly at random from $\{0,1\}^d$, and then draw $n$ i.i.d. domains from $\mathcal{P}_b$ to form a training set $G$. Let $\widehat{h}$ be the output of algorithm $A$ given $G$ as input. With probability at least $1 - \delta$, the output $\widehat{h}$ satisfies

$$\Pr_{\mathcal{D} \sim \mathcal{P}_b}[\mathrm{err}_{\mathcal{D}}(\widehat{h}) > \tau'] \leq \gamma. \tag{18}$$

Let $m$ be the number of domains among $\overline{\mathcal{D}}_1, \ldots, \overline{\mathcal{D}}_d$ that do not appear in the training set $S$. By Claim B.2, conditioned on $m$, with probability at least $1/2$, at least $m/2$ of these domains $\overline{\mathcal{D}}_i$ satisfy

$$\mathrm{err}_{\overline{\mathcal{D}}_i}(\widehat{h}) > \tau'.$$

That is, with probability at least $1/2$,

$$\Pr_{\mathcal{D} \sim \mathcal{P}_b}[\mathrm{err}_{\mathcal{D}}(\widehat{h}) > \tau'] \geq (m/2) \cdot (4\gamma/d) = (2m/d) \cdot \gamma.$$

Therefore, conditioned on $m > d/2$, with probability at least $1/2$, (18) does not hold. Since (18) holds with probability at least $1 - \delta$, we have

$$\Pr[m > d/2] \leq 2\delta,$$

or equivalently,

$$\Pr[d - m < d/2] \leq 2\delta. \tag{19}$$

The nonnegative random variable $d - m$ is the number of domains among $\overline{\mathcal{D}}_1, \ldots, \overline{\mathcal{D}}_d$ that *do* appear in the training set $S$. Since the training set $S$ is formed by $n$ i.i.d. examples drawn from $\mathcal{P}_b$ which puts $4\gamma$ total probability mass on $\overline{\mathcal{D}}_1, \ldots, \overline{\mathcal{D}}_d$, we have $\mathbb{E}[d - m] \leq 4\gamma n$. By Markov's inequality,

$$\Pr[d - m \leq 8\gamma n] \geq 1/2. \tag{20}$$

Combining (19) and (20) with our assumption that $\delta < 1/4$, we have

$$8\gamma n \geq d/2,$$

which implies that $\gamma \geq d/(16n)$. Moreover, with probability $(1 - 4\gamma)^n$, the training set $S$ contains no domains among $\overline{\mathcal{D}}_1, \ldots, \overline{\mathcal{D}}_d$, giving $d - m = 0$. Therefore, by (19),

$$2\delta \geq \Pr[d - m < d/2] \geq \Pr[d - m = 0] = (1 - 4\gamma)^n.$$

This implies that $\gamma \geq \Omega(\log(1/\delta)/n)$. In summary,

$$\gamma \geq \max\left\{ \frac{d}{16n}, \Omega\left( \frac{\log(1/\delta)}{n} \right) \right\} = \Omega\left( \frac{d + \log(1/\delta)}{n} \right). \qquad \square$$

## B.3  Proof of Theorem 5.1

**Theorem 5.1.** *For every positive integer d, there exists a hypothesis class $\mathcal{H}$ with $\mathrm{VCdim}(\mathcal{H}) = d$ satisfying the following property. For any $\alpha \in (0, 1/12)$, there exist $k = \Omega(d \log(1/\alpha))$ domains $\mathcal{D}_1, \ldots, \mathcal{D}_k$ such that*
$$\mathrm{Gdim}(\mathcal{H}, \{\mathcal{D}_1, \ldots, \mathcal{D}_k\}, 0.3, \alpha) = k.$$

We need the following helper lemma.

**Lemma B.3.** *For an odd positive integer $m$, on domain $X = \{0, 1, \ldots, m\}$, consider the hypothesis class $\mathcal{H} = \{h_1, \ldots, h_m\}$ where $h_i(x) = \mathbb{I}[x \geq i]$ for every $i = 1, \ldots, m$ and $x \in X$. There exists a distribution $\mathcal{D}$ of $(x, y) \in X \times \{0, 1\}$ such that*
$$\mathrm{err}_{\mathcal{D}}(h_i) = \begin{cases} 0.3 - \frac{1}{4m}, & \text{if } i \text{ is odd}; \\ 0.3 + \frac{1}{4m}, & \text{if } i \text{ is even}. \end{cases}$$

*Proof.* We construct the distribution $\mathcal{D}$ of $(x, y)$ as follows. With probability $1/2$, we choose $x = 0$. With the remaining probability $1/2$, we choose $x$ uniformly at random from $1, \ldots, m$. That is, $\Pr[x = i] = 1/(2m)$ for every $i = 1, \ldots, m$. Conditioned on $x = 0$, we draw $y \in \{0, 1\}$ such that $\Pr[y = 1 | x = 0] = 0.1$. Conditioned on $x \neq 0$, we choose $y = 1$ (deterministically) if $x$ is odd, and $y = 0$ if $x$ is even.

For odd $i \in \{1, \ldots, m\}$, we have

$$\mathrm{err}_{\mathcal{D}}(h_i) = \frac{1}{2} \cdot 0.1 + \frac{1}{2m} \sum_{x=1}^{m} \mathbb{I}[h_i(x) \neq \mathbb{I}[x \text{ is odd}]]$$

$$= 0.05 + \frac{1}{2m} \sum_{x=1}^{m} (\mathbb{I}[x < i \text{ and } x \text{ is odd}] + \mathbb{I}[x \geq i \text{ and } x \text{ is even}])$$

$$= 0.05 + \frac{1}{2m} \left( \frac{i-1}{2} + \frac{m-i}{2} \right)$$

$$= 0.3 - \frac{1}{4m}.$$

Similarly, for even $i \in \{1, \ldots, m\}$, we have

$$\mathrm{err}_{\mathcal{D}}(h_i) = 0.05 + \frac{1}{2m} \sum_{x=1}^{m} (\mathbb{I}[x < i \text{ and } x \text{ is odd}] + \mathbb{I}[x \geq i \text{ and } x \text{ is even}])$$

$$= 0.05 + \frac{1}{2m} \left( \frac{i}{2} + \frac{m-i+1}{2} \right)$$

$$= 0.3 + \frac{1}{4m}. \qquad \square$$

*Proof of Theorem 5.1.* We first prove the theorem in the special case where $d = 1$. On domain $X := \mathbf{Z}_{\geq 0} = \{0, 1, \ldots\}$, consider the hypothesis class $\mathcal{H} = \{h_1, h_2, \ldots\}$, where $h_i(x) = \mathbb{I}[x \geq i]$ for every $i = 1, 2, \ldots$ and $x \in X$. Clearly, $\mathrm{VCdim}(\mathcal{H}) = 1$.

Let $k$ be the largest integer satisfying $2^{k+2} + 4 < 1/\alpha$. By our assumption that $\alpha \in (0, 1/12)$, we have $k = \Omega(\log(1/\alpha))$. Our goal is to construct $k$ distributions $\mathcal{D}_1, \ldots, \mathcal{D}_k$ that are shattered by $h_1, \ldots, h_K$ for $K = 2^k$. Let $E_1, \ldots, E_K$ be all the $K$ subsets of $\{1, \ldots, k\}$. For every $j = 1, \ldots, k$, we will construct $\mathcal{D}_j$ such that for every $i = 1, \ldots, K$,

$$\begin{aligned} \mathrm{err}_{\mathcal{D}_j}(h_i) &< 0.3 - \alpha, & \text{if } j \in E_i; \\ \mathrm{err}_{\mathcal{D}_j}(h_i) &> 0.3, & \text{if } j \notin E_i. \end{aligned} \tag{21}$$

This ensures that $\mathcal{D}_1, \ldots, \mathcal{D}_k$ are shattered by $h_1, \ldots, h_K$, as needed to prove the lemma.

It remains to show the construction of $\mathcal{D}_j$ for every $j = 1, \ldots, k$. It will become convenient to choose $E_1$ to be the whole set $E_1 = \{1, \ldots, k\}$. This ensures that $j \in E_1$ for every $j = 1, \ldots, k$. For a fixed $j$, we partition $\{1, \ldots, K\}$ into consecutive non-empty blocks $S_1, \ldots, S_m$ for some $m \leq K$. Each

block $S_\ell$ has the form $S_\ell = \{x_{\ell-1} + 1, x_{\ell-1} + 2, \ldots, x_\ell\}$, where $0 = x_0 < x_1 < \cdots < x_m = K$. We define these blocks $S_1, \ldots, S_m$ so that

$$j \in E_i \quad \text{for every odd } \ell \in \{1, \ldots, m\} \text{ and every } i \in S_\ell;$$
$$j \notin E_i \quad \text{for every even } \ell \in \{1, \ldots, m\} \text{ and every } i \in S_\ell. \tag{22}$$

If $m$ is odd, we define $X' = \{x_0, x_1, \ldots, x_m\}$. If $m$ is even, we define $X' = \{x_0, x_1, \ldots, x_m, K + 1\}$. By Lemma B.3, there exists a distribution $\mathcal{D}_j$ over $X' \times \{0, 1\}$ such that

$$\text{err}_{\mathcal{D}_j}(h_{x_\ell}) \leq 0.3 - \frac{1}{4(m+1)} < 0.3 - \alpha \quad \text{for odd } \ell \in \{1, \ldots, m\},$$

$$\text{err}_{\mathcal{D}_j}(h_{x_\ell}) \geq 0.3 + \frac{1}{4(m+1)} > 0.3 \qquad \text{for even } \ell \in \{1, \ldots, m\}, \tag{23}$$

where we used $4(m + 1) \leq 4(K + 1) = 2^{k+2} + 4 < 1/\alpha$. Recall that for each $\ell = 1, \ldots, m$, the largest element in block $S_\ell$ is $x_\ell$, so for every $i \in S_\ell$,

$$h_i(x) = h_{x_\ell}(x) \quad \text{for every } x \in X'.$$

Therefore, (23) implies

$$\text{err}_{\mathcal{D}_j}(h_i) < 0.3 - \alpha \quad \text{for every odd } \ell \in \{1, \ldots, m\} \text{ and every } i \in S_\ell;$$
$$\text{err}_{\mathcal{D}_j}(h_i) > 0.3 \qquad \text{for every even } \ell \in \{1, \ldots, m\} \text{ and every } i \in S_\ell. \tag{24}$$

Combining (22) and (24) proves (21).

Finally, we prove the theorem for a general positive integer $d$. On input space $X' := [d] \times X$, we construct a hypothesis class $\mathcal{H}'$ using our hypothesis class $\mathcal{H}$ above as follows:

$$\mathcal{H}' := \{h : X' \to \{0, 1\} | \exists h_1, \ldots, h_d \in \mathcal{H} \text{ s.t. } h'(i, x) = h_i(x) \text{ for every } (i, x) \in X'\}.$$

This construction ensures that $\text{VCdim}(\mathcal{H}') = d \cdot \text{VCdim}(\mathcal{H}) = d$. By our analysis above, there exist $k = \Omega(\log(1/\alpha))$ domains $\mathcal{D}_1, \ldots, \mathcal{D}_k$ on $X$ that are shattered by $\mathcal{H}$. Now for every $i \in [d]$ and $j \in [k]$, we define $\mathcal{D}_{i,j}$ to be the distribution of $(i, x) \in X'$ with $x \sim \mathcal{D}_j$. These $kd = \Omega(d \log(1/\alpha))$ domains $(\mathcal{D}_{i,j})_{i \in [d], j \in [k]}$ are shattered by $\mathcal{H}'$, completing the proof. $\qquad\square$

### B.4 Proof of Theorem 5.2

**Theorem 5.2.** *Let $\mathcal{H}$ be an arbitrary hypothesis class with VC dimension $d$. For any set $\mathcal{G}$ of domains, any threshold $\tau \in \mathbb{R}$, and any margin $\alpha \in (0, 1/2)$,*

$$\text{Gdim}(\mathcal{H}, \mathcal{G}, \tau, \alpha) = O(d \log(1/\alpha)). \tag{11}$$

We will use the following classic result:

**Theorem B.4** (Sauer-Shelah-Perles Lemma [Sauer, 1972, Shelah, 1972])**.** *Let $\mathcal{F}$ be a class of concepts $f : Z \to \{0, 1\}$ defined on an arbitrary domain $Z$ with VC dimension $d$. Let $S$ be a subset of $Z$ with size $|S| = n \geq d$. Then*

$$|\mathcal{F}_S| \leq (en/d)^d,$$

*where $\mathcal{F}_S$ is the restriction of $\mathcal{F}$ to the subset $S \subseteq Z$.*

*Proof of Theorem 5.2.* Let $\mathcal{D}_1, \ldots, \mathcal{D}_k \in \mathcal{G}$ be $k$ domains shattered by $\mathcal{H}$. By the definition of shattering, there exist $K = 2^k$ hypotheses $h_1, \ldots, h_K \in \mathcal{H}$ with the following property: for every pair of hypotheses $h_{j_1}, h_{j_2}$ with $1 \leq j_1 < j_2 \leq K$, there exists $i \in \{1, \ldots, k\}$ such that

$$|\text{err}_{\mathcal{D}_i}(h_{j_1}) - \text{err}_{\mathcal{D}_i}(h_{j_2})| > \alpha. \tag{25}$$

Let us consider a fixed domain $\mathcal{D}_i$ for an arbitrary $i = 1, \ldots, k$. We show that for some $m = O(\alpha^{-2}k)$, there exists a dataset $S_i$ consisting of $m$ points $(x_1, y_1), \ldots, (x_m, y_m)$ such that

$$|\text{err}_{\mathcal{D}_i}(h_j) - \text{err}_{S_i}(h_j)| < \alpha/2 \quad \text{for every } j = 1, \ldots, K, \tag{26}$$

where $\text{err}_{S_i}(h_j)$ is the empirical error on $S_i$:

$$\text{err}_{S_i}(h_j) = \frac{1}{m} \sum_{\ell=1}^{m} \mathbb{I}[y_\ell \neq h_j(x_\ell)].$$

That is, $S_i$ is a representative dataset for distribution $\mathcal{D}_i$ in terms of measuring the error of the $K$ shattering hypotheses. We prove the existence of $S_i$ by the probabilistic method. For i.i.d. points $(x_1, y_1), \ldots, (x_m, y_m)$ drawn from $\mathcal{D}_i$, by the Chernoff bound, for a fixed $j$, (26) holds with probability at least $1 - 1/(2K)$ as long as $m \geq C\alpha^{-2}k$ for a sufficiently large absolute constant $C > 0$. By the union bound over $j = 1, \ldots, K$, with probability at least $1/2$, the $m$ data points satisfy (26), proving the existence of $S_i$.

We have now proved that for every $i = 1, \ldots, k$, there exists a size-$m$ dataset $S_i$ satisfying (26), where $m = O(\alpha^{-2}k)$. Combining this with (25), we know that for any pair of hypotheses $h_{j_1}, h_{j_2}$ with $1 \leq j_1 < j_2 \leq K$, there exists $i \in \{1, \ldots, k\}$ such that

$$\mathrm{err}_{S_i}(h_{j_1}) \neq \mathrm{err}_{S_i}(h_{j_2}).$$

This implies that $h_{j_1}(x) \neq h_{j_2}(x)$ for some $(x, y) \in S_i \subseteq (S_1 \cup \cdots \cup S_k)$. By Theorem B.4, for

$$n := |S_1 \cup \cdots \cup S_k| \leq mk = O(\alpha^{-2}k^2), \tag{27}$$

it holds that

$$2^k \leq (2 + en/d)^d. \tag{28}$$

Plugging (27) into (28) and taking the logarithm of both sides, we get

$$k/d \leq O(\log(1/\alpha) + \log(2 + k/d)).$$

This implies $k/d = O(\log(1/\alpha))$, proving (11). $\qquad\square$