# OpenReview forum: "How Many Domains Suffice for Domain Generalization? A Tight Characterization via the Domain Shattering Dimension"
_NeurIPS.cc/2025/Conference — NeurIPS 2025 poster_

### Official Review · Reviewer_M2uG · 2025-06-28

**Clarity:** 3
**Significance:** 2
**Originality:** 4
**Rating:** 3
**Confidence:** 1

**Summary:**

This paper investigates how many domains are needed to train a model that generalizes well to unseen domains from a theoretical perspective? This problem is formulated as a PAC learning setting and using a new combinatorial complexity measure called the domain shattering dimension, extensions analogous to classic PAC learning are offered. The domain shattering dimension captures how complex a hypothesis class is in relation to a family of domains, and it determines the number of domains required to ensure generalization. The central insight is that while VC dimension explains learning on a single domain, domain sample complexity quantifies how many domains a learner must see to perform well on unseen domains. The domain shattering dimension measures how well a hypothesis class can separate domains with respect to error thresholds. The authors provide tight upper and lower bounds on domain sample complexity using this new dimension. They design a min-max ERM algorithm and prove that the number of required domains depends logarithmically on the domain shattering dimension. Moreover, they show this measure is tightly related to the VC dimension. The paper also explores connections to domain adaptation, showing that when domains are similar (as measured by a variant of H-divergence), the domain shattering dimension is small.

**Questions:**

1. While this work might be an interesting work for a venue like COLT, it will not be very interesting in the absence of any empirical validation for the general audience of NeurIPS. Why there is not empirical experiments to validate the theoretical results or at least show their applicability in some cases?

2. What are the benefits of this work and the results for modern ML models?

3. The theory assumes the hypothesis class H is given and contains a suitable hypothesis, but how can this be verified in practice?

**Ethical Concerns:**

["NO or VERY MINOR ethics concerns only"]

**Final Justification:**

I acknowledge the theoretical merits of this work but remain unconvinced about practical benefits of this work and its applicability for deep learning. Given that the reviewers are divided, I leave the decision to AC and would be OK if my opinion is discarded in favor of positive opinions. I would also like to add that theoretical ML is not my research focus and I might have not fully grapsed the theoretical novelty of this work.

**Limitations:**

I would like to highlight that while I am familiar with the broad concepts of PAC-learnability and understand the paper, theoretical ML is not my research area and it is not something that I work about on a day to day basis. For this reason, my confidence in rating  is low because it is challenging for me to judge the novelty level of this work from theoretical perspective. I hope the rest of reviewers can compensate for it. Despite this shortcoming, I am quite familiar with empirical aspects domain generalization and domain adaptation.

1. Similar to PAC-learning, this work provides guarantees when the hypothesis class has finite domain sample complexity. Deep neural networks, however, typically have extremely large or even infinite VC dimension, making the standard PAC bounds uninformative in this context. As a result, the sample complexity guarantees become unhelpful.

2.  The theory is built around ERM. In deep learning, however, practical optimization cannot search the entire hypothesis space and often only finds local minima, not the global one required by ERM. This means the theoretical guarantees do not directly translate to the way deep learning models are actually trained.

3. While deep learning models can be trained efficiently in practice, the theoretical guarantees about computational tractability provided by PAC-learning are not tight or meaningful for modern neural architectures.

4. The classic PAC model assumes the data can be generated by some function within the hypothesis class which is rarely true for deep learning.

**Quality:**

3

**Strengths And Weaknesses:**

Strengths:

- The paper is well written and can be followed without the need to check the details of proofs.

- Introduction of domain shattering dimension for characterizing domain generalization sample complexity is novel and new.

- The results and algorithms apply to binary classification, multi-class classification, and regression.

Weaknesses:

- The practical value of this work to the current trend in AI is unclear.

- There is no empirical verification.

- The domain shattering dimension is hard or even impossible to compute or estimate for practical deep networks, similar to how VC dimension is often intractable.

- A core assumption is that the hypothesis class contains a single model that performs well across all domains. But in diverse domain distributions. this assumption is probably wrong.

-

---

> ### Author Rebuttal · Authors · 2025-07-30
>
> Thank you for your careful review and thoughtful feedback! We will make sure to revise our paper based on your comments. Below are our responses to your specific points.
>
> **Regarding experiments and practical benefits.** We believe there **is** a non-trivial audience at NeurIPS interested in foundational advances in theoretical machine learning. While our work is purely theoretical, we’d like to emphasize two key takeaways:
>
> 1. **Conceptual insight**: We introduce a new framework for analyzing domain generalization, which avoids the need to explicitly model domain similarity—something that is often difficult in practice.
>
> 2. **Algorithmic contribution**: We also provide a practical algorithmic principle—**min-max ERM**—which can be implemented efficiently using standard optimization techniques such as gradient descent or multiplicative weights for concrete hypothesis classes such as neural networks or linear models. Our analysis suggests that min-max ERM is provably more robust than standard ERM in multi-domain settings.
>
> We hope these insights can inform the design and analysis of future algorithms, and potentially guide empirical work moving forward. We will highlight this in our revision.
>
> **Regarding the assumption that H contains a suitable hypothesis.** Unlike the realizability assumption in classical PAC learning, we do **not** assume the existence of a perfect hypothesis in $\mathcal{H}$. We only assume that there exists a **reasonably good** hypothesis $h^\*$ that performs well (i.e. has reasonably small error) across domains—analogous to how one might expect a CNN to generalize reasonably well across diverse image domains (e.g., natural images, medical images, etc.). We evaluate the performance of the learned model $\hat h$  by comparing it with this reasonably good hypothesis $h^\*$ (similarly to the **agnostic** setting of PAC learning).
>
> Of course, if $\mathcal{H}$ does not contain such a hypothesis $h^\*$—just as a CNN might fail on certain image types—then the issue is not with the theory, but with the choice of hypothesis class. In such cases, one would need to explore a richer or more appropriate class (e.g., a new architecture). How to **select or learn** an appropriate hypothesis class for domain generalization is an important and open direction for future work.
>
>
>
> **Regarding dimension of deep neural networks.** We’d like to clarify that our proposed dimension can, in fact, be small even for deep neural networks when the domains are similar. Consider $\mathcal{H}$ as the class of all networks defined by varying parameters within a fixed architecture. In the extreme case where all domains are identical, the domain shattering dimension $\text{Gdim}(\mathcal{H}, \mathcal{G}, \tau, \alpha)$ becomes 0, regardless of the VC dimension of the network.
>
> We agree that in the classical PAC setting, VC dimension leads to vacuous bounds for deep nets because it accounts for the worst-case over all distributions. However, in our framework, the domain space is taken as an input, allowing the dimension to reflect structure in the distribution over domains. As a result, the domain sample complexity can remain meaningful even for expressive hypothesis classes like neural networks. That said, estimating this dimension in practice—especially for deep models—is still an open and challenging question, which we view as an interesting direction for future work.
>
> **Regarding local minima versus global minima.** Our analysis does not require solving ERM exactly. For any threshold $\tau$, if the algorithm returns a hypothesis that achieves error at most $\tau$ on each observed domain, then our generalization bound still applies. In practice, this means that approximate solutions—such as those obtained via standard deep learning optimization methods—can still yield meaningful guarantees, as long as the per-domain training errors remain controlled. Running exact ERM would simply allow for a smaller achievable $\tau$, but it is not necessary for our theoretical results to hold. We will include this discussion in our revision.

---

### Official Review · Reviewer_BJVM · 2025-07-01

**Clarity:** 2
**Significance:** 2
**Originality:** 3
**Rating:** 4
**Confidence:** 2

**Summary:**

The paper studies an important problem in domain generalization(DG): how many randomly sampled domains are needed to perform well on unseen domains? To this end, the papers develops a new measure, domain shattering dimension and also establish a relationship between this dimension and the classic VC dimension.

**Questions:**

The following are the questions/suggestions:

1. The motivation lines in 71-74 need detailed justification.

2. It is not sure how will the developed theory be used in practical scenarios? Atleast a toy experiment is needed to establish the significance of the developed theory.

3. The connection to domain adaptation section is not well explained. How will the relaxed H-divergence upper bound the target error? Can the same idea be extended to H \delta H divergence? This divergence is actually used in the final upper bound developed in Ben-David et. al [2010]

**Ethical Concerns:**

["NO or VERY MINOR ethics concerns only"]

**Final Justification:**

The rebuttal has addressed most of my concerns.

**Limitations:**

Yes

**Quality:**

3

**Strengths And Weaknesses:**

Strengths:

1. The paper addresses an important problem that is highly relevant in practical scenarios.

2. It is always valuable to see theory in DG: the paper develops a new combinatorial measure characterizing domain sample complexity and establishes its relationship with VC dimension.

Weaknesses:

1. Weak motivation: In Lines 71–74, the paper claims that for a different threshold \tau' the fat shattering dimension grows large, but provides no detailed justification for this assertion.

2. Missing practical experiment: Seminal theoretical papers in domain adaptation and domain generalization (Ben-David et al., 2010; Blanchard et al., 2011) always accompany their theory with a practical experiment demonstrating its usefulness. This paper provides no such experiment.

3. The section 6, where the paper connects the developed domain shattering dimension to domain adaptation is not explained clearly. In lines 312-314, it is mentioned that the domain shattering dimension upper bounds the error. But in the paper it only establishes the relationship between relaxed H-divergence and H-divergence in Line 321. It is not clear if this relaxed H-divergence can upper bound the target error? Also in Ben-David et al. [2010] the upper bound that is finally developed uses H \delta H divergence. Will a similar relaxed version of H \delta H divergence exist?

4. There are several typographical errors: Lines 143, 213, 313 (it should be domain adaptation and not domain generalization), 81 (before this line there is no reference or explanation about what is partial concept class) and 165 (\gamma is not defined)

---

> ### Author Rebuttal · Authors · 2025-07-30
>
> Thank you for your careful reading and thoughtful review! We will make sure to revise our paper based on your comments and correct the typos you pointed out. Below are our responses to your specific points.
>
> **Regarding Lines 71-74.** We will revise our paper to include the definition of the classical fat-shattering dimension and refine these lines to make them clearer. The main message is that the classical fat-shattering dimension is not an appropriate complexity measure in our setting because it is **maximized over all choices of thresholds** $\tau\in [0,1]$. For instance, let us fix $\tau = 0.3$. Consider a hypothesis class $\mathcal{H}$ where every $h \in \mathcal{H}$ satisfies $\text{err}_D(h) \leq 0.3$ for all domains $D$. Moreover, for any subset of domains $E$, there exists an $h \in \mathcal{H}$ such that $\text{err}_D(h) \leq 0.2$ if and only if $D \in E$. In this case, domain generalization with respect to $\tau = 0.3$ is trivial, yet the fat-shattering dimension is large (because it is large at another threshold $\tau’ = 0.2$). We hope this clarifies Lines 71-74. Please let us know if you have further questions.
>
> Our domain shattering dimension can be viewed as a variant of the fat-shattering dimension with a fixed threshold $\tau$. This modification makes the characterization tight, which we prove using many new ideas beyond the classic theories about the fat-shattering dimension (e.g. Lemma 4.2).
>
> **Regarding experiments.** While our work is purely theoretical, we’d like to emphasize two key takeaways:
>
> 1. **Conceptual insight**:
> Our results show that it is possible to analyze domain generalization *without explicitly modeling domain similarity*. This is insightful because domain similarity is often difficult to model in practice. The proposed domain shattering dimension captures inter-domain structure *implicitly*, offering a flexible and general framework.
>
> 2. **Algorithmic contribution**: We also provide a practical algorithmic principle—**min-max ERM**—which can be implemented efficiently using standard optimization techniques such as gradient descent or multiplicative weights for concrete hypothesis classes.
>
> We believe these contributions are valuable in their own right and will highlight them in the revised paper. While experimental validation is important, toy experiments are unlikely to offer insights beyond what our theory already captures. Designing meaningful empirical studies that reflect real-world scenarios while providing additional insight beyond our theoretical results remains an open and important question.
>
>
> **Regarding the section on connection to domain adaptation.** Thank you for the comment! We will refine the presentation of this section to make it clearer.
>
> Our Theorem 6.1 shows that the covering size $|\mathcal G'|$ under the $(\mathcal{H},\tau)$-divergence gives an upper bound on the domain shattering dimension. Combining this with our main upper bound (Theorem 4.1), we get an upper bound on the target error in terms of the covering size $|\mathcal G'|$ under the $(\mathcal{H},\tau)$-divergence.
>
> While Theorem 6.1 is stated (and proved in Appendix B.5) for the $(\mathcal{H},\tau)$-divergence, it applies to the $\mathcal{H}$-divergence as well because the latter only makes the covering size become larger (or remain the same). This follows from the basic fact that the $(\mathcal{H},\tau)$-divergence never exceeds the $\mathcal{H}$-divergence. We will make this clearer in the revision.
>
> We don't know if the $\mathcal H \Delta \mathcal H$-divergence has a relaxed form (as you suggest) that would meaningfully bound the target error. However, our result extends to the original $\mathcal H\Delta \mathcal H$-divergence as defined in Ben-David et. al [2010]. Lemma 4 of Ben-David et. al [2010] proves an upper bound on the $\mathcal H$-divergence (defined in our paper at Line 318 as the maximum error difference over $h\in \mathcal H$) using the $\mathcal H\Delta \mathcal H$-divergence. Combining that with our Theorem 6.1 gives an upper bound on the domain shattering dimension using the $\mathcal H\Delta \mathcal H$-divergence, and by Thm 4.1 we get an upper bound on the target error using the $\mathcal H\Delta \mathcal H$-divergence. Thank you for bringing the $\mathcal H\Delta \mathcal H$-divergence to our attention. We will include this result in our revision.
>
> **Typographical Errors.**  Thank you for finding these! We will correct them in the revision.

---

> > ### Comment · Reviewer_BJVM · 2025-08-04
> > **Response**
> >
> > Thank you for your detailed response. Most of my concerns have been addressed. I will increase my score accordingly.

---

### Official Review · Reviewer_3Zud · 2025-07-03

**Clarity:** 3
**Significance:** 2
**Originality:** 3
**Rating:** 4
**Confidence:** 3

**Summary:**

This paper tackles a "meta-learning" problem of understanding domain generalization in the same vein as how sample complexities are derived in PAC learning to give statistical guarantees. In this setting, there exists a family of domains $\mathcal{G}$, where every domain is a distribution on instance-label pairs, and a concept class $\mathcal{H}$ and the goal is to characterize the number of domains needed to be observed in the training data such that it is possible to select an $h \in H$ that yields low error on unseen domains. The idea is that there is a distribution $\mathcal{P}$ over $\mathcal{G}$ and using PAC-like statistical guarantees, to produce a classifier $h$, with high probability, that has low expected error on all domains in $\mathcal{G}$ with respect to $\mathcal{P}$.

The authors construct a new combinatorial quantity called the domain shattering dimension that is defined in somewhat similar fashion to the fat shattering dimension. It essentially measures the "capacity" of $\mathcal{G}$ in the sense that it captures the largest set $S \subseteq \mathcal{G}$ such that for any subset $S' \subseteq S$, there exists a classifier $h$ with error below a certain threshold on domains in $S'$ and error above a threshold for all domains in $S \setminus S'$. The authors use the domain shattering dimension to construct both upper and lower bounds on the number of domains needed to achieve successful domain generalization and also explore connections with the VC-dimension.

**Questions:**

- If my understanding is correct, is it true that the VC dimension of the partial concept class generated from $\mathcal{G}$ is equivalent to the domain shattering dimension?
- This is a follow-up to my point in the weakness section. Is there a somewhat altered notion of the domain shattering dimension that additionally takes into account the structure of domains in $\mathcal{G}$ with respect to the instance space? To me (I could be wrong), it seems to make sense that the domain sample complexity can also be tied to the inherent structure of the domains itself because one could leverage this information in learning regardless of the different domains seen by the learner.
- In $(\tau, \alpha, \gamma, \delta)$-domain learnability, is there a size on what $m$ is required to be? Is the size of $m$ inferred from VC theory on the number of samples needed to obtain uniform convergence guarantees on $\mathcal{H}$?

**Ethical Concerns:**

["NO or VERY MINOR ethics concerns only"]

**Final Justification:**

Overall, I lean positively toward this paper getting accepted at NeurIPS. My primary concern with this paper was their lower bound construction in showing that the domain shattering dimension is critical when observing the sample complexity. The authors successfully addressed my concerns and I recommend the authors to add a more detailed explanation of the lower bound construction in that section of the paper.

**Limitations:**

Yes

**Quality:**

3

**Strengths And Weaknesses:**

Strengths:
- The paper tackles a new problem of domain generalization that hasn't been explored before.
- The paper is quite well-written and the technical details of their constructions are accessible.
- For the upper bound, the technique of casting the domain learnability problem as a partial concept class and then using machinery established by Alon et al. to produce a uniform convergence guarantee is quite clever.

Weaknesses:
- My primary concern is the way the lower bound is constructed to show that the sample complexity using the domain shattering dimension is indeed tight up to logarithmic factors. As explained in the first paragraph of Section 4.2, the domain shattering dimension doesn't necessarily provide a lower bound for the domain sample complexity in all cases. For the specific mentioned by authors in Section 4.2, it seems that even the domain sample complexity generated from the domain shattering dimension provides a pretty loose upper bound (i.e. much larger than the true domain sample complexity). Generally, one tries to show the validity of a combinatorial quantity by showing that it measures the hardness of a learning instance through lower and upper bounds that work in the same exact learning setting. Here, the family $\mathcal{G}$ is augmented to the family of $\mathcal{G}'$ to prevent certain degenerate cases and the lower bound is on this $\mathcal{G}'$. Altering the learning setting for the lower bound seems a bit concerning because the lower bound no longer certifies the hardness of the original problem instance.

---

> ### Author Rebuttal · Authors · 2025-07-30
>
> Thank you for appreciating our work and providing insightful comments. Below are our answers to your specific questions:
>
> **Is the VC dimension of the partial concept class equivalent to the domain shattering dimension?** Yes. This is mentioned at Line 213 of the paper. We can make it more visible by upgrading it to a Lemma or a Claim in the revision. We appreciate your suggestions.
>
> **Regarding the lower bound construction and the idea of an altered notion of the domain shattering dimension.** Your idea makes perfect sense to us, and we appreciate your thoughtful comment. The gap between our upper and lower bounds arises because the min-max ERM algorithm we use for the upper bound does not require any knowledge of the domain space $\mathcal{G}$ beyond the observed domains—whereas, in principle, there exist domain families $\mathcal{G}$ that are trivially easy to learn if such knowledge were available. Proving a tight upper bound for every choice of $(\mathcal H, \mathcal G)$ without the $\mathcal G'$ augmentation requires a more advanced algorithm that leverages the inherent structure of $\mathcal G$. This is an intriguing follow-up question of our work and we will mention it explicitly in the revision.
>
> However, in many practical settings we have little knowledge about the inherent structure of $\mathcal{G}$, and this is a scenario where the min-max ERM algorithm is well suited for.  (This lack of knowledge is certainly the case for the real-life medical research problem that originally piqued our interest in the problem, where at training time we have little knowledge about the space of patient distributions of unseen hospitals in the future.) An alternative way to interpret our lower bound is as a universal hardness result: there always exists a slightly extended domain space $\mathcal{G}'$ for which the lower bound applies and the min-max ERM algorithm is (near) optimal. Our current lower bound construction explicitly provides such a $\mathcal{G}'$.
>
> **In the learnability definition, is there a size requirement on m?** In the learnability definition, we only require m to be finite. In the algorithm and upper bound, we just adopt $m = O(\frac{\text{VCdim}}{\epsilon^2})$ from classical VC theory (see Lines 186-188).

---

> > ### Comment · Reviewer_3Zud · 2025-08-04
> >
> > I thank the authors for taking the time to address my review.
> >
> > The authors have successfully addressed all my questions regarding their paper. Overall, I retain my score of a 4 for this paper and recommend the authors to add the paragraph on the lower bound construction in their revision.

---

### Official Review · Reviewer_KZw4 · 2025-07-05

**Clarity:** 4
**Significance:** 4
**Originality:** 4
**Rating:** 5
**Confidence:** 4

**Summary:**

The paper studies domain adaptation, focusing on quantifying how many domains are sufficient to learn a model that performs well across both seen and unseen domains within a family. Under the assumption that there exists an estimator that achieves low error across all domains, the authors derive upper and lower bounds on the domain sample complexity, introducing a new combinatorial quantity, the "domain shattering dimension." The upper bound is established on Min-Max ERM.
Interestingly, the authors also construct counterexamples demonstrating that this bound is not instance-optimal, indicating that further improvements may be possible in specific settings. Additionally, the paper draws connections between the proposed framework and existing measures in domain adaptation, such as the H-divergence.

**Questions:**

Personally, I would have also been interested in connections to the underlying data generating process, e.g., whether causal assumptions could help (sparse mechanism shift; independent causal mechanisms) - cf. https://arxiv.org/abs/1206.6471, https://proceedings.mlr.press/v48/gong16.html, https://ojs.aaai.org/index.php/AAAI/article/view/9542. But don't take this as a criticism of the paper.

**Ethical Concerns:**

["NO or VERY MINOR ethics concerns only"]

**Limitations:**

Limitations are discussed in the last section.

**Quality:**

4

**Strengths And Weaknesses:**

This is a serious theoretical contribution. The authors formalize the domain adaptation problem in a rather general setting—without assumptions on domain similarity—and provide tight upper and lower bounds applicable to arbitrary hypothesis classes and classes of domains.
The results connect the new notion of domain shattering dimension with established concepts from the literature, offering a unified perspective on domain generalization and adaptation.
One potential drawback is that, in practice, it can be challenging to clearly specify or identify the class of domains relevant to a given problem. Computing the domain shattering dimension would then be difficult.

---

> ### Author Rebuttal · Authors · 2025-07-30
>
> Thank you for appreciating our work! Also thank you for pointing us to those papers—it's definitely interesting to explore how our framework might connect to data-generating processes modeled through causal assumptions.

---

> > ### Comment · Reviewer_KZw4 · 2025-08-06
> > **Response**
> >
> > I have read the response and the other reviews, and would still like to see this paper at the conference.

---

### Official Review · Reviewer_VFhB · 2025-07-06

**Clarity:** 2
**Significance:** 1
**Originality:** 2
**Rating:** 3
**Confidence:** 3

**Summary:**

This paper analyses the domain sample complexity—that is, how many domains must be randomly sampled to learn a model that performs reliably on both seen and unseen domains from a given family—within the PAC learning framework. Specifically, the authors derive both upper and lower bounds on the domain sample complexity using a novel combinatorial measure they introduce, called the domain shattering dimension. Furthermore, the authors establish a tight quantitative relationship between the domain shattering dimension and the classical VC dimension. This connection implies that any hypothesis class that is learnable under the standard PAC framework is also learnable under their proposed setting.

**Questions:**

Please see the strengths and weaknesses.

**Ethical Concerns:**

["NO or VERY MINOR ethics concerns only"]

**Final Justification:**

**Note to Authors:**
The proof shows that the min-max ERM algorithm, which takes the maximum over sampled domains, is a consequence of the min-max objective in Definition 3.1. If the objective were defined using the standard ERM approach, the resulting algorithm would be an ERM algorithm. Therefore, the conclusion that the theory suggests the min-max ERM algorithm is superior to ERM is not true.

**Overall:**
The paper treats domains as individual samples and defines a new loss function at the domain level. This loss function is then compared to the 0/1 loss on samples using a threshold $\tau$. The paper applies sample complexity bounds similarly at the domain level. The current results suggest that with a sufficiently large number of training domains (assuming the domains are iid sampled), generalization is guaranteed. However, this formulation does not appear to offer substantial insights for designing algorithms for domain generalization.

**Limitations:**

Yes

**Quality:**

2

**Strengths And Weaknesses:**

## Strengths:

The authors derive both upper and lower bounds on domain sample complexity using a combinatorial measure they introduce, termed the domain shattering dimension, which appears to offer some degree of novelty, although its practical or theoretical implications are not immediately clear.

## Weaknesses:

- The paper does not provide any clear practical or theoretical implications for the domain generalization (DG) problem. In DG, we typically assume access to a limited number of source domains, and the key challenge is how to leverage these to generalize to unseen target domains. A more relevant formulation would focus on to what extent a model trained on a small number of domains can generalize well. In contrast, the question this paper raises—how many domains are needed to guarantee generalization to any unseen domain—feels somewhat unrealistic and disconnected from the practical challenges in DG. This formulation does not appear to offer useful insights for designing algorithms for DG.
- Moreover, the authors seem to treat domains as individual samples and then define a new loss function at the domain level, comparing it to 0/1 loss on samples via a threshold $\tau$. They then apply sample complexity bounds analogously at the domain level. However, this domain complexity upper bound does not seem to provide any additional insight beyond standard sample complexity bounds in the context of DG.
- The paper states: "domain sample complexity can still be very small if the domains in \mathcal{G} are very similar." However, domain similarity is difficult to quantify, and even if we had a metric, it would only apply to the observed training domains. This observation is only meaningful if the number of training domains is large and diverse enough to represent a meaningful meta-distribution. If the training domains are limited and highly similar, it's unclear what conclusion we can draw—do we need more diverse training domains, or have we already captured enough information?
- Regarding the connection to domain adaptation, the authors suggest defining a new divergence measure. However, it is unclear what the benefit of this new divergence is, or how it contributes to our understanding or improvement of domain adaptation techniques. This section lacks a clear motivation and purpose.

---

> ### Author Rebuttal · Authors · 2025-07-30
>
> Thank you for your careful review and thoughtful feedback! We will make sure to revise the paper based on your comments. Below are our responses to your specific points.
>
> **Regarding practical and theoretical implications / insights.** Before we make a clarification about your comment on the problem formulation, we’d like to highlight that our proposed min-max ERM is a **practical algorithmic principle** that can be implemented using standard optimization techniques—for example, gradient descent, multiplicative weights, or other iterative methods—for concrete hypothesis classes like neural networks or linear models. Please keep in mind that our goal is to characterize **how many *observed* source domains are sufficient** for learning a good model (hypothesis) that **generalizes well to the *unseen* target domains.**
>
> **Regarding the problem formulation you suggest.** The formulation you suggest is, in fact, what we study in our paper. While we formulate the question as “how many observed domains are sufficient” to generalize to $(1-\epsilon)$ fraction of the unseen target domains, this question can be equivalently formulated as “how many unseen domains” one can generalize to given data from a fixed, small number $n$ of observed domains. For example, a result saying that “$n = 1/\epsilon$ observed domains are sufficient to achieve generalization on $(1-\epsilon)$ fraction of the unseen domains” is equivalent to the result that “one can generalize to $(1 - 1/n)$ fraction of the unseen domains given data from $n$ observed domains”. While the two formulations are equivalent, the latter formulation is indeed more natural and relevant in practical scenarios, where the number of observed domains is limited and not controlled by the learner (as you pointed out).  **In fact, our main theorems (Theorems 4.1 and 4.3) are stated using this more relevant formulation,** where we express $\epsilon$ (the fraction of unseen domains that we *don’t* generalize to, denoted by $\text{Er}_{\mathcal P,\tau}(\hat h)$ in the paper) as a function of the number $n$ of observed domains. We hope this clarifies the implications of our results to DG practice and will include this clarification in the revised paper.
>
> **Regarding the treatment of domains as individual samples.** We believe that treating domains as individual samples *is* a key conceptual contribution of our work. Prior theoretical work typically assumes strong structural relationships across domains, such as similarity under a predefined divergence measure. However, in real-world applications, it is often unclear how to define or justify such similarity measures. This motivated us to ask whether domain generalization is possible **without making explicit structural assumptions**. Our framework addresses this by borrowing tools from statistical learning theory and analyzing generalization across domains in a similar way PAC theory analyzes generalization across examples. A major advantage of our approach is that it allows us to characterize domain generalization using our domain shattering dimension, which **does not require any predefined similarity measure.** As we show in Section 6, when the domains *are* similar under classical measures (e.g., the H-divergence of Ben-David et al.), our domain shattering dimension becomes small—implying that it naturally captures domain similarity *without requiring it to be modeled explicitly*.
>
> While our proof technique involves a reduction to binary labels, it is not trivial to use such a reduction to prove a tight characterization with matching upper and lower bounds. We prove our tight characterization using many novel components in addition to the reduction. For example, if one defines a binary loss at the domain level (e.g., via a fixed threshold $\tau$), the setting may resemble standard PAC learning, but this does not lead to a tight upper bound. To ensure tightness, we need two thresholds $\tau - \alpha$ and $\tau$ and map values between them to a special label $\bot$ beyond the binary labels 0 and 1. To analyze the *partial* hypotheses obtained from this reduction, we leverage deep results (Theorem B.1) from Alon et al. [2022] to prove a novel uniform convergence bound (Lemma 4.2) beyond the scope of Alon et al. [2022] (see Footnote 1 on Page 6).
>
> **Regarding domain similarity.** We appreciate the reviewer’s concern. To clarify: the sentence "domain sample complexity can still be very small if the domains in $\mathcal{G}$ are very similar" aims to discuss the property of domain sample complexity instead of describing an observation. It refers to a **hypothetical extreme scenario** where all domains in $\mathcal{G}$ are highly similar (or even identical). In such cases, the domain sample complexity can indeed be small—we mention this in the introduction to emphasize that domain sample complexity depends not only on the hypothesis class $\mathcal{H}$, but also on the domain family $\mathcal G$. We did **not** intend to suggest that such similarity can be inferred from a small number of training domains. We will make this clearer in the revision.
>
> **Regarding the section on domain adaptation and divergence measure.** The goal of this section is to connect our framework to prior work, specifically to show that we can **recover insights from Ben-David et al. [2010]** within our setting and thereby demonstrate the **expressiveness** of our approach.
>
> Ben-David et al. define the $\mathcal{H}$-divergence to measure similarity between domains, and show that generalization is possible when the source and target domains are close under this measure. This leads to a **clustering interpretation:** if the domain space admits a small cover under $\mathcal{H}$-divergence, then observing one domain per cluster suffices.
>
> We show that our **domain shattering dimension implicitly captures this structure.** Specifically, a small domain shattering dimension is guaranteed by a small covering number under our new divergence, which is in turn guaranteed by a small cover under the $\mathcal{H}$-divergence. The takeaway is that our framework captures domain similarity **without requiring it to be explicitly modeled,** yet can still recover and generalize known results from domain adaptation—highlighting its power and flexibility. We will emphasize this point as the motivation of the section in the revised paper.
>
> The new divergence measure is introduced solely as an intermediate step to bridge our shattering-based analysis with the divergence-based analysis. We will make this clearer in the revision.

---

> > ### Comment · Reviewer_VFhB · 2025-08-06
> >
> > I appreciate the authors' rebuttal. However, my concerns still remain. Specifically, in the context of DG, where there are a limited number of source domains, the objective is to train on these sources to generalize effectively to unseen target domains.
> >
> > **Practical and Theoretical Implications:**
> >
> > A DG algorithm should focus on leveraging the given source domains to generalize to target domains, using techniques such as invariant features, augmentation, causal inference, etc. This work investigates how many domains are needed to achieve effective generalization. My question is how the theoretical results from this study can contribute to the design of DG algorithms, as I cannot identify any clear practical or theoretical implications for that.
> >
> > **Domain Shattering Dimension and Similarity Measure:**
> >
> >  In a scenario with a limited number of source domains, my concern is that even with a solid and implementable metric, it can only be applied to the source domains themselves. We cannot conclusively determine the characteristics of the domain space based solely on the similarity between a small number of source domains. This is why I mentioned that the metric would only be meaningful when the number of source domains is sufficiently large to adequately represent the entire domain space.
> > A potential issue is when the source domains are highly similar to each other, but the target domain differs. This scenario is common in DG benchmarks like DomainBed, where this could lead to false conclusions.
> >
> > **Connecting to Domain Adaptation (DA):**
> >
> > Given that source and target domains often appear different in the input space (which is why DA is needed), the practical implication of H-divergence is that it minimizes the distance between the source domain representation and the target domain representation, enabling generalization from source domains to target domains. Does this new divergence measure offer a similar practical implication? If not, how is it intended to be used?

---

> > > ### Author Response · Authors · 2025-08-06
> > >
> > > Thanks for your response and follow-up questions.
> > >
> > > > Practical and Theoretical Implications:
> > > A DG algorithm should focus on leveraging the given source domains to generalize to target domains, using techniques such as invariant features, augmentation, causal inference, etc. This work investigates how many domains are needed to achieve effective generalization. My question is how the theoretical results from this study can contribute to the design of DG algorithms, as I cannot identify any clear practical or theoretical implications for that.
> > >
> > > The main algorithmic takeaway is that **min-max ERM**—which minimizes the maximum loss across source domains—can achieve optimality in our theoretical framework. This suggests that, instead of standard ERM (which minimizes average loss over pooled data across source domains), **min-max ERM is a more principled choice for domain generalization**.
> > >
> > >
> > > > Domain Shattering Dimension and Similarity Measure:
> > > In a scenario with a limited number of source domains, my concern is that even with a solid and implementable metric, it can only be applied to the source domains themselves. We cannot conclusively determine the characteristics of the domain space based solely on the similarity between a small number of source domains. This is why I mentioned that the metric would only be meaningful when the number of source domains is sufficiently large to adequately represent the entire domain space. A potential issue is when the source domains are highly similar to each other, but the target domain differs. This scenario is common in DG benchmarks like DomainBed, where this could lead to false conclusions.
> > >
> > >
> > > We fully agree with your concern: measuring similarity based on a small number of source domains may not reflect the full structure of the domain space. However, we want to clarify that our framework **does not** require modeling or defining a similarity measure, **nor** does it rely on similarities among a small number of source domains to represent the domain space.
> > >
> > > Instead, the **domain shattering dimension** captures the underlying diversity of the domain space directly, **without relying on any explicit similarity metric**.
> > > - If the domain space is **very diverse**, the shattering dimension is **large**, indicating that more source domains are needed for generalization.
> > > - If the domain space is **less diverse**, the shattering dimension is **small**, and only a few source domains may suffice.
> > >
> > > This avoids the pitfall of inferring structure from limited domain similarity and provides a principled theoretical tool that adapts to the true diversity of the domain space.
> > >
> > >
> > >
> > >
> > >
> > > > Connecting to Domain Adaptation (DA):
> > > Given that source and target domains often appear different in the input space (which is why DA is needed), the practical implication of H-divergence is that it minimizes the distance between the source domain representation and the target domain representation, enabling generalization from source domains to target domains. Does this new divergence measure offer a similar practical implication? If not, how is it intended to be used?
> > >
> > > Yes, our divergence measure has a similar practical implication. In some sense, it can be viewed as a **refined version of H-divergence** that focuses on hypotheses which perform well on at least one domain in the domain space. These are the only hypotheses that matter in practice, since models that perform poorly on all domains will never be selected.
> > >
> > > Because our divergence only considers this relevant subset of hypotheses, it is often **smaller than classical H-divergence**.

---

> > > > ### Comment · Reviewer_VFhB · 2025-08-09
> > > >
> > > > > The main algorithmic takeaway is that **min-max ERM**
> > > >
> > > > This follows naturally from your problem setting (Definition 3.1: Optimal domain error bound and optimal hypothesis), rather than from new theoretical results. It is a similar setup to the worst-case generalization framework, as in [1] and related follow-up work.
> > > > From the theory, the only conclusion I can currently see is that if domains are sampled i.i.d. from the domain distribution, then having more domains generally leads to better performance. However, I believe this scenario is not very practical.
> > > >
> > > > **Domain Shattering Dimension:**
> > > >
> > > > 1. The statement “The domain shattering dimension captures the underlying diversity of the domain space directly” is still unclear to me. In practice, how can we determine the level of diversity in the domain space when we only have access to a limited set of domains?
> > > > 2. Even if we could reliably estimate this diversity, the resulting conclusion seems binary: either “we need more domains” or “we already have enough.” In that case, what specific advantages does this offer for the actual design of DG algorithms?
> > > >
> > > > **Connecting to Domain Adaptation (DA):**
> > > >
> > > > This concern has been addressed. I will raise my rating to 3.
> > > >
> > > > [1] Sagawa, Shiori, et al. "Distributionally robust neural networks for group shifts: On the importance of regularization for worst-case generalization." ICLR 2020.

---

### Note · Authors · 2025-08-12

We thank all reviewers for their thoughtful comments and appreciate their engagement in the discussion. We appreciate all reviewers’ decision to either increase their score or maintain their positive score. We also appreciate that Reviewer VFhB agrees to increase their score from 2 to 3 (though we note that the score currently shown on the site is still 2).


As implied in Reviewer VFhB’s response, we have addressed their concerns regarding problem formulation/ treatment of domains as individual samples/connection to domain adaptation. Due to the timing of their last response, we didn’t have a chance to address their follow-up questions on the practical implications of max-min ERM and Domain Shattering Dimension during the discussion period. We would like to take this opportunity to clarify these points.

### **Final response to Reviewer VFhB**

First, we’d like to emphasize that we **prove** the optimality of min-max ERM--something that does not follow directly from Definition 3.1. In Definition 3.1, the min-max objective is defined with the maximum taken over all domains. In contrast, the min-max ERM algorithm takes the maximum over sampled domains. We provide a generalization bound and establish its optimality via a matching lower bound. We believe this theoretical contribution should not be overlooked.

Regarding the **domain shattering dimension**, we agree that it may be difficult to compute in practice. However, it is not required to run the algorithm. Instead, it serves as a theoretical tool for analyzing the optimality of min-max ERM. Specifically, we prove an upper bound in Theorem 4.1 and a matching lower bound in Theorem 4.3—both with similar dependence on this dimension—which together establish optimality. We also compare it to the VC dimension to highlight the gap in learning difficulty between standard statistical learning and domain generalization.

We acknowledge that this dimension is a foundational theoretical contribution and may not yield immediate practical insights. However, it plays a crucial role in capturing the learnability of domain generalization.

---

### Decision · Program_Chairs · 2025-09-17

**Decision:**

Accept (poster)

**Comment:**

[nice summary lifted from one of the reviews] The paper studies domain adaptation, focusing on quantifying how many domains are sufficient to learn a model that performs well across both seen and unseen domains within a family. Under the assumption that there exists an estimator that achieves low error across all domains, the authors derive upper and lower bounds on the domain sample complexity, introducing a new combinatorial quantity, the "domain shattering dimension."

The theoretically-inclined reviewers seem to praise the results of the paper (5-4-4). They also engaged in discussion with the authors, who clarified most issues that were raised. On the other hand the more skeptical reviewers insist more on limited applicability issues in practice (3-3). Given the strongly theoretical/fundamental inclination of the contribution and the novel theoretical contributions pointed out by the reviewers, I am in favor of accepting.